



**A CONCEPTUAL MODEL-BASED SEDIMENT CONNECTIVITY ASSESSMENT FOR**
**PATCHY AGRICULTURAL CATCHMENTS**
Pedro V. G. Batista[1*], Peter Fiener[2], Simon Scheper[1], Christine Alewell[1]
[1]Department of Environmental Sciences, Universität Basel, Bernoullistrasse 30, 4056, Basel,
Switzerland.
[2]Institute for Geography, Universität Augsburg, Alter Postweg 118, D – 86159, Augsburg, Germany.
[*]pedro.batista@unibas.ch





**ABSTRACT**
The accelerated sediment supply from agricultural soils to riverine and lacustrine environments leads to
negative off-site consequences. In particular, the sediment connectivity from agricultural land to surface
waters is strongly affected by landscape patchiness and the linear structures that separate field parcels
(e.g. roads, tracks, hedges, and grass-buffer-strips). Understanding the feedbacks between these
structures and sediment transfer is therefore crucial for minimising off-site erosion impacts. Although
soil erosion models can be used to understand lateral sediment transport patterns, model-based
connectivity assessments are hindered by the uncertainty in model structures and input data. In
particular, the representation of linear landscape features in numerical soil redistribution models is often
compromised by the spatial resolution of the input data and the quality of the process descriptions. Here
we adapted the WaTEM/SEDEM model using high resolution spatial data (2 m x 2 m) to analyse the
sediment connectivity in a very patchy mesoscale catchment (73 km$^2$) of the Swiss Plateau. Specifically,
we used a global sensitivity analysis to explore model structural assumptions about how linear landscape
features (dis)connect the sediment cascade. Furthermore, we compared model simulations of hillslope
sediment yields from five sub-catchments to tributary sediment loads, which were calculated with long-
term water discharge and suspended sediment measurements. Our results showed that roads were the
main regulators of sediment connectivity in the catchment. In particular, the sensitivity analysis revealed
that the assumptions about how the road network (dis)connects the sediment transfer from field-blocks
to water courses had a much higher impact on modelled sediment yields than the uncertainty in model
parameters. Moreover, model simulations showed a higher agreement with tributary sediment loads
when the road network was assumed to directly connect sediments from hillslopes to water courses. Our
results ultimately illustrate how a high-density road network combined with an effective drainage system
increase sediment connectivity from hillslopes to surface waters in this representative catchment of the
Swiss Plateau. This further highlights the importance of considering linear structures in soil erosion and
sediment connectivity models.



## 1 INTRODUCTION

Rainfall events on sloped surfaces continuously displace small amounts of soil, which are transported downslope as sediments. These sediments are then stored and remobilised several times before conceivably reaching surface waters. Accordingly, the sediment cascade is a natural and potentially long geomorphological process (Fryirs, 2013). However, the accelerated sediment supply from agricultural soils to riverine and lacustrine environments leads to negative off-site consequences. Specifically, nutrient-rich and pollutant-bound particulate matter from arable land is associated to the eutrophication and contamination of water courses (Krasa et al., 2019; Laceby et al., 2021). Extreme erosion events in agricultural fields are also linked to the occurrence of muddy floods (Boardman, 2020) and to damages to downstream infra-structure (Bauer et al., 2019). Therefore, understanding how and when sediment is transferred from agricultural fields to different landscape compartments is imperative to reduce off-site erosion impacts.

The degree with which a system facilitates sediment transfer within its internal compartments is defined by Heckmann et al. (2018) as sediment connectivity. This concept can be further distinguished into a structural component, associated to the semi-static spatial configuration of the landscape; and a functional one, which emerges as a dynamic property of the hydro-sedimentological system (Wainwright et al., 2011). Connectivity theory therefore provides a framework to rethink the sediment delivery problem (Fryirs, 2013; Parsons et al., 2009) and to understand the complex spatio-temporal processes that regulate sediment transport.

In agricultural landscapes, sediment connectivity is strongly affected by the patchiness of the land use configuration, and the presence of linear features between field parcels (e.g. hedges, grass-buffer-strips, and roads) (Alder et al., 2015; Bakker et al., 2008; Chartin et al., 2013; Fiener et al., 2011; Van Oost et al., 2000). The importance of landscape patchiness in regulating sediment transfer is specifically relevant in areas where a large number of small fields, separated by linear structures, create a complex hydrological system. However, the experimental analysis of sediment connectivity at catchment scale is challenging, as it involves measuring both internal soil redistribution processes and cascading sediment transport rates. The interaction between landscape patchiness, linear structures, and sediment connectivity is therefore not addressed by the typical setup of experimental erosion studies, which either focus on small erosion plots or catchment sediment yields (Fiener et al., 2019).

Due to the difficulties in measuring the processes that affect sediment movement at catchment and landscape scale, it is common practice to analyse connectivity with modelling approaches (Nunes et al., 2018). These usually rely on high-resolution process-based models, assuming they are able to explicitly take connectivity into account (Baartman et al., 2020); semi-qualitative indices (Borselli et al., 2008; Cavalli et al., 2013); or more recently, the coupling of conceptual models with probability theory (Mahoney et al., 2020a, 2020b). In specific, the use of process-based soil erosion and sediment transport models might be an important pathway to improve our understanding of sediment connectivity (Nunes





et al., 2018). However, erosion models in general, and process-based models in particular, face two
fundamental problems for representing sediment connectivity : (i) the input data requirements are large
and uncertain, and model application is often restricted to small catchments with a few square kilometres
(e.g. Baartman et al., 2020; Starkloff and Stolte, 2014; Wilken et al., 2017) and (ii) the implemented
process descriptions, especially along linear landscape features and field boundaries, are weekly defined
due to the aforementioned unavailability of experimental data. On the other hand, Borrelli et al. (2018)
demonstrated how parcel-specific high resolution land cover and management data can improve soil
erosion/sediment delivery models in patchy agricultural catchments.
Here, we aimed to (i) adapt a conceptual soil erosion and sediment delivery model with high spatial
resolution data (2 m x 2 m) within a Monte Carlo framework; (ii) to analyse the sediment connectivity
in a very patchy mesoscale catchment (73 km²) in Switzerland; and (iii) to perform a sensitivity analysis
of model parameters and structural assumptions regarding how linear features (dis)connect the sediment
cascade. Hence, we demonstrate how models can be used to understand the interaction between linear
features, landscape patchiness, and sediment connectivity. This will contribute to increase our
comprehension of relevant connectivity processes and our ability to develop appropriate measures for
reducing off-site erosion impacts.
**2 MATERIALS AND METHODS**
**2.1 Study catchment**
The study catchment consists of the contributing area of the Baldegg Lake, in the central Swiss Plateau
(Figure 1). The lake has been extensively studied due to its hypertrophic waters, which have been
artificially oxygenated since 1983 (e.g. Lavrieux et al., 2019; Müller et al., 2014; Teranes and
Bernasconi, 2005). The eutrophication of the lake has been mostly linked to excessive phosphorus loads
during the 20th century (Wehrli et al., 1997). Although water quality in the lake is currently improving
(BAFU, 2016), the supply of phosphorus-rich sediment is still a concern to local authorities (Stoll et al.,
2019). Hence, we chose to focus our study on the Baldegg catchment by reason of the ongoing research
in the area and the availability of comprehensive hydrological data, which has been monitored by the
department of environment and energy of Canton Luzern. Importantly, the catchment is representative
of the patchy agricultural landscape of the Swiss Plateau, as we detail below.




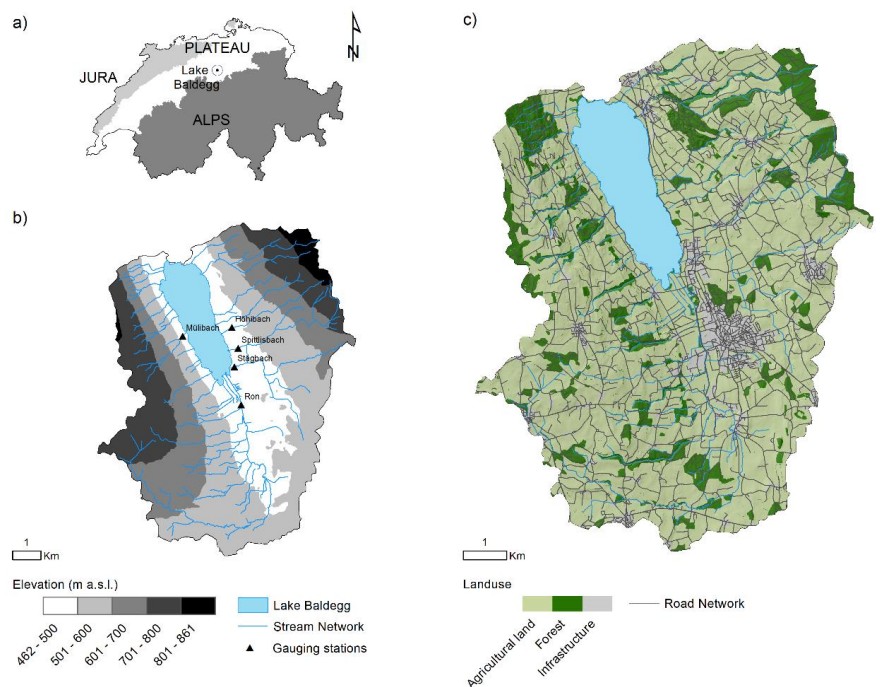

Figure 1. a) Location of the Lake Baldegg catchment; b) elevation, stream network, and location of
hydrological gauging stations; c) land use. Data source: Swisstopo, 2020.

The Baldegg catchment has a total area of 73.2 km², of which 5.2 km² are covered by the lake. The
remaining area is occupied by agricultural land (74%), forests (16%), and settlements (10%) (Swisstopo,
2020) (Figure 1c). The agriculture consists of intensively managed temporary pastures, cereal
production under crop rotation, permanent grasslands, fruit orchards, and small vineyards (Lavrieux et
al., 2019; Stoll et al., 2019). Agricultural field-blocks, here delimited by external boundaries (e.g. roads,
water courses, and forests) (Bircher et al., 2019), have a median size of 4.4 ha. However, smaller patches
separated by hedges, tree lines, and grass-buffer-strips, are generally found within the blocks (Figure 2).

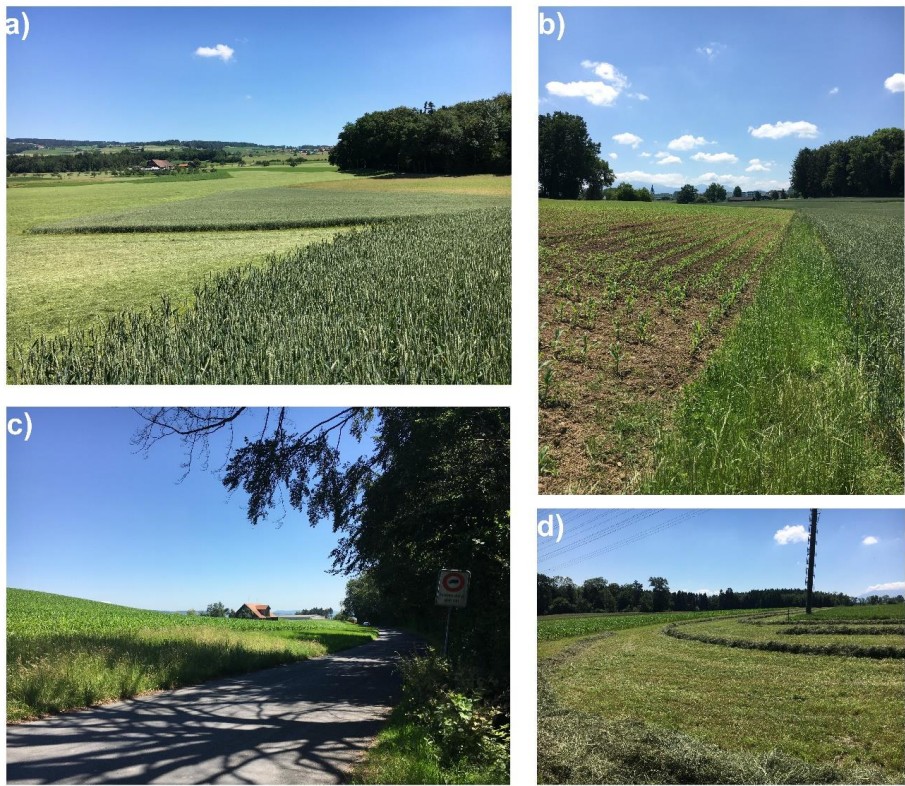

109

Figure 2. Typical agricultural landscapes from the Baldegg catchment: a) Small arable and grassland
patches within larger field-blocks, b) Grass-buffer-strip between maize and wheat fields, c) wide grass-
buffer-strip between maize field and a vicinal road, d) freshly cut hay from a pasture in between maize
fields.

The road network density in the Baldegg catchment is 6.0 km km$^{-2}$, which is approximately three times
higher than the stream density (1.9 km km$^{-2}$). Streams in the upper catchment are often incised, with
visible, yet not prominent, signs of bank erosion. Flow is sometimes regulated in the lowland areas, and
tile drainage is found at water accumulation zones (Stoll et al., 2019). A total of 22 channels flow into
the Baldegg Lake, of which five streams are monitored for water and sediment discharge by cantonal
authorities, as described in section 2.2.

Elevation in the Baldegg catchment ranges from 462 to 861 m.a.s.l. Steeper slopes (maximum 35°) and
higher altitudes are found in the eastern and western sides of the catchment (Figure 1b), in a typical
glacial landscape of the Swiss Plateau – in this case formed by the retreat of the Reuss Glacier in the
south to north direction (~18,000 years BP) (Keller, 2021; Pfiffner, 2021). As a result, calcaric



Cambisols developed upon Tertiary and Quaternary deposits are the main soil class in the catchment.
Rainfall is well distributed throughout the year, although greater precipitation is observed from May to
August. The average annual rainfall (2010-2020) at the closest gauging station is ~ 1000 mm yr$^{-1}$
(Mosen, 454 m a.s.l., ~3.5 km north of the Baldegg lake, acquired from MeteoSwiss) and mean rainfall
erosivity in the catchment is ~ 1150 MJ mm ha$^{-1}$ h$^{-1}$ yr$^{-1}$ (Schmidt et al., 2016).
**2.2 Tributary suspended sediment loads**
Suspended sediment concentrations from five tributaries flowing into the Baldegg Lake were measured
during ten years (Jan 2010 - Dec 2019) by the Department of Environment and Energy of Canton Luzern.
Approximately 275 grab samples were taken from each tributary, which corresponds roughly to two
samples per month, and high-flow events were opportunistically sampled (Figure 3). Suspended
sediments were measured at the same location where water discharge was monitored by automatic
gauging stations (Figure 1b). A summary of the measured rainfall, water discharge, and sediment
concentration from 2010 to 2020 is displayed in Figure 3.



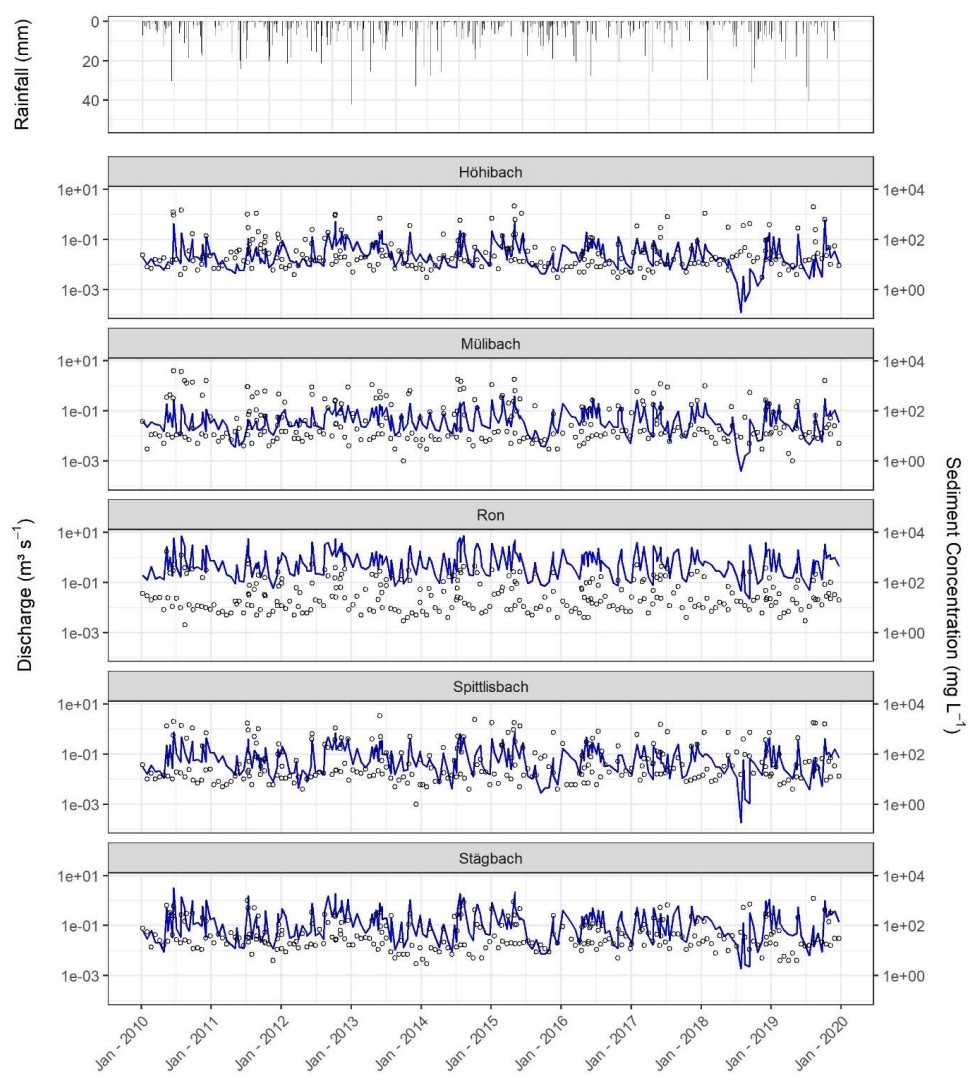


Figure 3. Daily rainfall at the Mosen station, mean daily discharge (blue line), and sediment
concentration (circles) at the monitored tributaries of the Baldegg Lake (2010-2019).



In order to calculate the sediment load for the monitored tributaries, we fitted a rating curve (Equation
1) with the measured sediment concentrations and their correspondent water discharge values.
Additional covariates were included to account for hysteresis, seasonality, and constituent exhaustion
(Table 1) (Vigiak and Bende-Michl, 2013; Wang et al., 2011):

$$\ln c_i = \beta_0 + \sum_{k=1}^{5} \beta_k x_{k,i} + \varepsilon_i \tag{1}$$


Where: $c$ is sediment concentration (mg L$^{-1}$) for day $i$, $\beta_0$ is the intercept, $\beta_k$ are fitted coefficients, $x_k$ are
covariates (Tab. 1) accounting for discharge, hysteresis, seasonality and constituent exhaustion, and $\varepsilon_i$
is the residual error.
Table 1. Covariates used for fitting the sediment-rating curve, as in Vigiak and Bende-Michl (2013) and
Wang et al. (2011).

| Covariate | Expression | Explanation | Physical interpretation |
|---|---|---|---|
| $x_{1,i}$ | $\ln Q_i$ | $Q_i$ is = water discharge for day $i$ (m$^3$s$^{-1}$) | Discharge |
| $x_{2,i}$ | $(\ln Q_i)^2$ | Quadratic term of $Q_i$ | Hysteresis |
| $x_{3,i}$ | $\sin(2\pi M_i/12)$ | $M_i$ = month of day $i$ | Seasonality |
| $x_{4,i}$ | $\cos(2\pi M_i/12)$ | $M_i$ = month of day $i$ | Seasonality |
| $x_{5,i}$ | $\dfrac{\sum_{z=1}^{i} 0.95^{i+1-z} Q_z}{\sum_{z=1}^{i} 0.95^{i+1-z}}$ | Discount flow up to day $i$ | Constituent exhaustion (see Wang et al., 2011) |


The rating curve was used to estimate daily sediment concentrations for the entire 2010-2020 period.
Subsequently, we propagated the uncertainty in the regression fit by simulating posterior distributions
of the model coefficients ($\beta_0$, $\beta_k$) with an informal Bayesian function of the R package '*arm*' (Gelman
and Hill, 2007), as in Batista et al. (2021). The posterior distributions were used to simulate 1000
sediment concentration values for each day $i$. These were transformed into daily distributions of
sediment loads (Mg), considering the mean daily discharge measurements from the gauging stations.
Sediment loads were ultimately aggregated into average annual values (Mg yr$^{-1}$).
**2.3 Model description**
A modified version of the spatially distributed erosion and sediment transport WaTEM/SEDEM (Van
Oost et al., 2000; Van Rompaey et al., 2001; Verstraeten et al., 2010) was used in this study.
WaTEM/SEDEM provides a framework for modelling sediment connectivity from hillslope to water
courses by use of a steady-state transport capacity equation and a pixel-based sediment routing
component. That is, the model assumes that soil particles displaced by water erosion at a given grid-cell



are transferred downstream for as long as the runoff transport capacity is greater than the sediment
supply, or until the flow path reaches a definite sink. Although the model is able to simulate both tillage
and water erosion, here we focus on the latter, which is calculated with an adaptation of the RUSLE
(Renard et al., 1997):

$$A = R\ K\ LS_{2d}\ C\ P \tag{2}$$


Where: $A$ is average annual soil loss (kg m$^{-2}$ yr$^{-1}$), $R$ is rainfall erosivity (MJ mm m$^{-2}$ h$^{-1}$ yr$^{-1}$), $K$ is soil
erodibility (kg h MJ$^{-1}$ mm$^{-1}$), $LS_{2d}$ is a topographic factor calculated by the Desmet and Govers (1996)
procedure (dimensionless), $C$ is a cover-management factor (dimensionless), and $P$ is a support practice
factor (dimensionless).
Transport capacity (kg m$^{-1}$ yr$^{-1}$) is assumed to be proportional to the potential to rill erosion, which is
described by a power function of slope length and gradient (Van Rompaey et al., 2001):

$$TC = K_{TC}RK(LS_{2d} - 4.12\ S_g^{0.8}) \tag{3}$$


Where: $K_{TC}$ is a landuse-dependent transport capacity coefficient (m) which requires calibration, R is
rainfall erosivity (MJ mm h$^{-1}$ yr$^{-1}$), K is soil erodibility (t h MJ$^{-1}$ mm$^{-1}$), $LS_{2d}$ is a topographic factor
calculated by the Desmet and Govers (1996) procedure (dimensionless), and $S_g$ is slope gradient (m m$^{-1}$

181    ).

WaTEM/SEDEM partially incorporates the influence of the landscape structure on sediment transfer by
the use of a parcel connectivity parameter $P_{Con}$, which represents the proportion of sediment that is
stopped at field borders. The model also simulates runoff connectivity by means of a parcel trapping
efficiency $P_{TEf}$ parameter, which corresponds to the proportion of the flow accumulation that is routed
downstream. Finally, the model is able to estimate the total amount of sediment transferred from
hillslopes to water courses, which can be interpreted as the hillslope component of a catchment sediment
budget. Since WaTEM/SEDEM does not represent channel erosion or in-stream deposition processes,
any comparison between modelled sediment yields and catchment-outlet sediment loads must be
interpreted with upmost caution. For further information on the model, we refer to Notebaert et al.,
(2006), Van Oost et al., (2000), Van Rompaey et al., (2001), and Verstraeten et al., (2010).
**2.4 Model implementation, input data, and sensitivity analysis**
WaTEM/SEDEM is usually implemented with a user-friendly GUI developed at KU Leuven, and freely
available at https://ees.kuleuven.be/geography/modelling/watemsedem/. Although the software





facilitates model application, it does not allow for more complex operations, such as sensitivity or
uncertainty analysis. Moreover, some model components might not be fully comprehensible without
access to the source-code, and WaTEM/SEDEM is frequently used as a black-box. Hence, in order to
perform a sensitivity analysis of model parameters and underlying structural model assumptions, we
implemented a WaTEM/SEDEM version using the free open source software R (R Core Team, 2021)
and SAGA GIS (Conrad et al., 2015). The main adaptations are described in the following, and our code
is available as supplementary material.
Our model application consists of a global all-at-a-time sensitivity analysis, as described by Pianosi et
al. (2016). That is, we performed a Monte Carlo simulation to explore the variability of the whole
parameter space, and all input factors were sampled simultaneously for each model realisation (n =
1200). The framework is similar to an uncertainty analysis, except in this case we did not focus on
quantifying uncertainty or locating the parameter space which produced behavioural model realisations.
Instead, we concentrated on apportioning sources of uncertainty to different model input factors (Pianosi
et al., 2016). This should allow us to identify parameters and model assumptions that have a greater
impact on the manner with which WaTEM/SEDEM describes sediment connectivity in the Baldegg
catchment.
For each iteration of the Monte Carlo simulation, all RUSLE input variables were sampled from uniform
distributions, except for the $LS_{2d}$ factor (Table 2). Minimum and maximum $R$ factor values were
retrieved from the Swiss national map (Schmidt et al., 2016), and a single lumped value for the whole
catchment was sampled for each iteration. The same approach was used for the $K$ factor (Schmidt et al.,
2018). We used lumped catchment values for these factors due to their low spatial variability within the
study area, according to the national maps (coefficient of variations are 1% and 7% for the $K$ and $R$
factor, respectively). For the $C$ and $P$ factors, here combined in a single $CP$ parameter, uniform
distributions were created for each landuse class in the catchment, based on commonly used values from
the literature and a rasterised (2 m x 2 m) land cover map (1:25000) (Swisstopo, 2020). Due to the
unavailability of spatially distributed crop statistics in the Baldegg catchment, pastures and cropland
were aggregated into a single arable land category (Table 3). In this case, minimum and maximum values
were relaxed to represent a wide possible combination of crops and support practices. Such
combinations were assessed with the $CP$-Tool (Kupferschmied, 2019), which allows for the calculation
of $CP$ values considering common crop rotation systems in Switzerland. Finally, the $LS_{2d}$ factor was
calculated with a slope (rad) and an upslope contributing area ($m^2$) grid, which were obtained by
processing a 2 m x 2 m resolution DEM from SwissALTI3D (Swisstopo, 2014).


Table 2. Minimum and maximum parameter values sampled during the Monte Carlo simulation.

| Parameter | Category | Min | Max |
|---|---|---|---|
| $R$ (MJ mm m$^{-2}$ h$^{-1}$ yr$^{-1}$) | | 950 10$^{-4}$ | 1350 10$^{-4}$ |
| $K$ (kg h MJ$^{-1}$ mm$^{-1}$) | | 0.025 10$^{3}$ | 0.040 10$^{3}$ |
| $CP$ | Arable land | 0.01 | 0.5 |
| | Grass-buffer-strips | 0.001 | 0.009 |
| | Forest | 0.0001 | 0.003 |
| | Orchard | 0.001 | 0.2 |
| | Vineyard | 0.05 | 0.6 |
| $K_{TC}$ (m) | High | 1 | 200 |
| | Low | 1 | 100 |
| $P_{TEf}$ | | 0 | 1 |
| $P_{Con}$ | | 0 | 1 |


Similarly, all WaTEM/SEDEM-specific model parameters were sampled from uniform distributions
(Table 2). Landuse classes with a $CP$ factor above 0.01 received higher transport capacity coefficients
($K_{TC}$ high). The remaining landuse classes, namely forests and grass strips, received lower coefficients
($K_{TC}$ low). The $K_{TC}$ reference values were taken from Van Rompaey et al. (2001) and extended in order
to explore a larger parameter space. The sampled parcel trapping efficiency ($P_{TEf}$) values were assigned
to forests and grass-buffer-strips in the rasterised land cover map. The resulting $P_{TEf}$ grid was used as a
weight for calculating the aforementioned upslope contributing area. Hence, only a proportion of the
grid-cell area from forests and grass-strips contributes to the downstream flow accumulation, as runoff
amounts are assumed not to increase (or to increase slowly) with slope length under natural vegetation
(Govers, 2011). Parcel connectivity ($P_{Con}$) values were assigned to the forest and grass-buffer-strips cells
that bordered agricultural fields. The transport capacity (Eq. 2) at these cells was reduced by a fraction
inversely proportional to the sampled $P_{Con}$ value.
For each sampled combination of parameters values, the models were ran with and without the presence
of grass-buffer-strips between agricultural field-blocks and adjacent roads and forests. Although grass-
buffer-strips are generally present at field borders in the Baldegg catchment (Figure 2), these features
were not distinguishable in the land cover map. Hence, we manually inserted 2 m wide grass-buffer-
strips at the aforementioned borders. The extent of the buffer-strips in reality is quite variable, and
generally wider at forest vicinities, as required by law in Switzerland (Alder et al., 2015). For simplicity,
we used a single value that should allow us to test the sensitivity of the model to the presence of the
strips. On the other hand, hedges and tree lines within field-blocks were already classified in the land
cover map, and required no additional processing.
Furthermore, three road connectivity assumptions were assessed for each model iteration. In a first
scenario, roads were treated as an ultimate sink, with zero transport capacity (i.e. 'roads as sinks').
Hence, sediments reaching roads or infrastructure were subsequently removed from the system and did



not reach surface waters. This represents a scenario in which road and field drainage traps most
sediments and partly diverges runoff to wastewater treatment plants. A second scenario assumed that all
sediments reaching the road network were directly connected to the stream network. This represents a
situation in which the drainage system acts as a hydrological shortcut, transferring sediments from fields
into surface waters (i.e. 'roads as shortcuts') (see Schönenberger and Stamm, 2020). As in the original
model formulations (see Notebaert et al., 2006), the third scenario assigned very high transport capacity
to roads and infrastructure, so that no deposition would take place (i.e. 'roads as patch-connectors'). In
this case, runoff and sediment might flow along or across the road network – which is expected to happen
during extreme rainfall events when the drainage system is clogged. Hence, sediment transfer will be
entirely dependent on the flow direction calculated from the DEM. Here we employed a multiple flow
direction algorithm, which was used for calculating upslope contributing area and routing sediments
along the flow-path. The sediment routing component was implemented with a capacity accumulation
function from SAGA GIS (Conrad et al., 2015). Of note, all geo-processing tools were applied with the
'RSAGA' package (Brenning et al., 2018). Additional R packages essential to the simulations were
'doParallel' (Ooi et al., 2019), 'foreach' (Calway and Weston, 2017), 'raster' (Hijmans, 2020), and
'rgdal' (Binvand et al., 2019).
The sensitivity of WaTEM/SEDEM to the uncertainty in model parameters, the presence of grass-buffer-
strips, and assumptions about road connectivity was assessed by evaluating modelled hillslope sediment
yields (i.e., the amount of sediment delivered from hillslopes to surface waters) for the entire Baldegg
catchment. A qualitative analysis was performed with a visual inspection of scatter plots, comparing the
univariate parameter space with the model response surface. Additionally, we used a random forest
analysis to rank the importance of input factors to the uncertainty in model outputs (Antoniadis et al.,
2021). That is, a random forest predicted modelled sediment yields based on the sampled parameter
values in the Monte Carlo simulation. The importance of the input factors, including model parameters,
the presence of grass-strips, and the road connectivity scenarios, was ranked based on their relative
contribution to the RFA predictive error, following an out-of-bag estimate (Breiman, 2001). We chose
the RFA due to its ability to rank both qualitative and quantitative input factors. The analysis was
performed with the 'randomForest' (Liaw and Wiener, 2002) R package.
Finally, we compared the resulting WaTEM/SEDEM simulations of sub-catchment hillslope sediment
yields to the suspended sediment loads from the monitored tributaries. Of note, with this comparison we
only aim to provide a general picture of the plausibility of the model realisations. Suspended sediment
loads are a product of a complex interaction of hillslope and channel remobilisation processes, which
are not represented by WaTEM/SEDEM. Hence, modelled hillslope yields and suspended loads are not
fully commensurable, and we did not focus on a rejectionist framework for model testing. This research
is exploratory, and investigates the importance of linear features and landscape patchiness on sediment
connectivity.



## 3 RESULTS

### 3.1 Sensitivity analysis

The road connectivity assumptions were by far the most sensitive input factor for WaTEM/SEDEM in the Baldegg catchment. This can be easily visualised in Figure 4, which presents scatter plots comparing sampled parameter values and the model response surface. The uniformly scattered points denote a low sensitivity of the modelled hillslope sediment yields to most input factors, with some evident exceptions: $CP$ for arable land, $K_{TC}$ high, and $K_{TC}$ low. On the other hand, all plots demonstrate that higher sediment yields were calculated when we assumed that roads behaved as hydrological shortcuts, directly connecting agricultural patches to the stream-network.



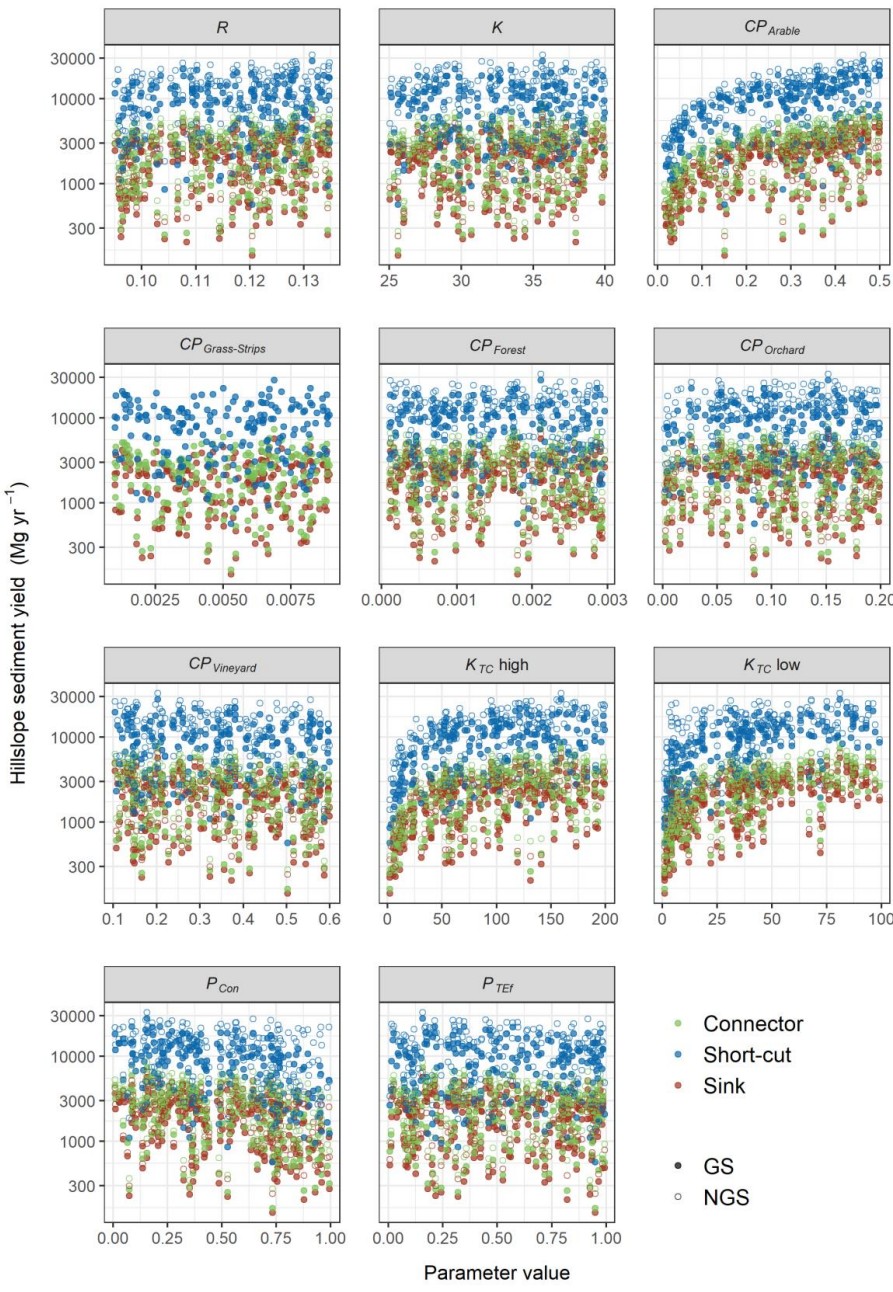

298

Figure 4. Univariate scatter plots of sampled parameter values. Full circles represent model realisations
with the presence of grass-buffer-strips (GS), and open circles represent the ones without strips (NGS).
Colours represent the road connectivity assumptions (i.e. 'roads as patch-connectors', 'roads as
hydrological shortcuts', and 'roads as sinks'). See section 2.4 for a description of road connectivity
scenarios.



Similarly, the results from the RFA demonstrate that road connectivity was the most important input
factor for predicting the WaTEM/SEDEM outputs (Figure 5). That is, if road connectivity was not
considered, the mean-squared-error (MSE) increased in 175%. The MSE increase associated to $CP$ for
arable land (67.3%), $K_{TC}$ low (35.6%), $K_{TC}$ high (34.3%), and the presence of grass-buffer-strips
(27.0%), indicate the model was also sensitive these input factors. However, if we considered each road
connectivity scenario individually, the results from the random forest were shifted, as the model seemed
to be more sensitive to the presence of grass-buffer-strips for the 'road as shortcuts' scenario (MSE
increase = 43.6%).

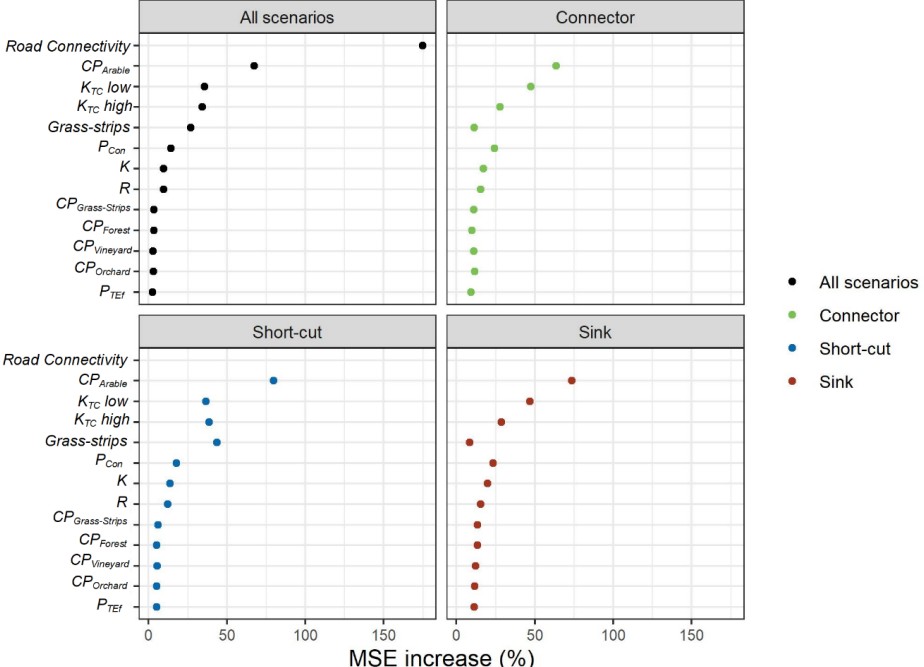


Figure 5. Mean-squared-error (MSE) increase associated to model input factors for the RFA. Larger
relative errors indicate the input factors were more important for estimating model outputs.
**3.2 Spatial patterns**
The spatial patterns of soil redistribution rates were also highly influenced by linear features, landscape
patchiness, and connectivity assumptions. Sediment deposition on field-blocks downslope from roads
was more frequently observed for the 'roads-as-connectors' scenario, than for the other road
connectivity assumptions. Specifically, when sediments were not diverged or trapped by the road
network, there was a greater proportion of sediment deposition on foot-slope field borders and other
potential sinks (Figure 6b).



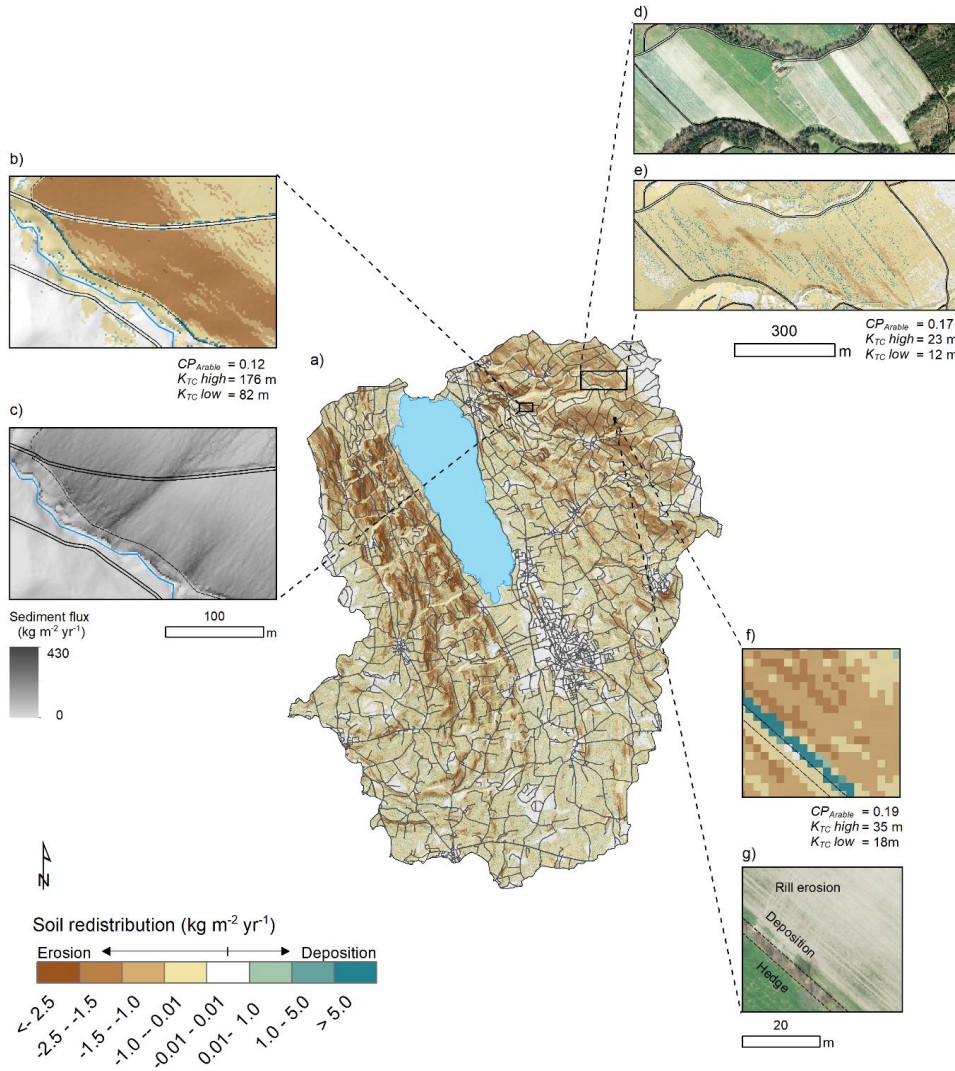


Figure 6. a) Catchment patterns of soil redistribution for model a realisation with the presence of grass-buffer-strips; b) detail of sediment deposition on field borders, 'road as patch connectors' scenario; c) detail of sediment fluxes across the road network, 'road as patch connectors' scenario'; d) detail of aerial image of multiple parcels within a field-block (SwissImage, 2014); e) soil redistribution rates for the field-block; f) detail of sediment deposition at a grass-buffer-strip at a field border; g) aerial image for the field (SwissImage, 2014).



The sediment flux from agricultural fields was generally interrupted when entering forest patches, and
further deposition was modelled at forested valley floors, near the stream channels, for all scenarios
(Figure 6b, c). Importantly, sediment deposition along grass-buffer-strips, hedges, and tree lines reduced
sediment fluxes in between field-blocks, forming a patchy connectivity pattern. This was again visible
for all simulated connectivity assumptions, albeit particularly pronounced when the presence of grass-
buffer-strips was considered (Figure 6 a, f).
Unexpectedly, the soil redistribution patterns revealed that WaTEM/SEDEM simulated linear
deposition areas at the borders of small cropland patches (Figure 6d, e). This occurred even in the
absence of grass-buffer-strips or hedges, and hence without $P_{Con}$ parameterisation, which was only
applied to field-block borders. These depositional patterns were particularly evident within field-blocks
oriented across the slope direction, and apparently stem from small-scale changes in the slope gradient,
which were represented by the high-resolution DEM and which potentially results from long-term tillage
erosion.

**3.3 Soil redistribution rates, hillslope sediment-yields, and suspended sediment loads**

Soil redistribution rates for eroding grid-cells in the Baldegg catchment were almost identical among
the simulated road connectivity assumptions (Table 3). Higher absolute deposition rates were calculated
for the simulations without grass-strips for both the connector and sink scenarios, which is a result of
increased erosion rates calculated without the presence of the strips. On the other hand, lower sediment
yields were calculated with the presence of grass-buffer-strips when the connectivity scenarios were
analysed individually. Among these scenarios, deposition rates were lower if roads were considered to
behave as hydrological shortcuts. Contrarily, deposition rates for the 'roads as connectors' and 'roads
as sinks' scenarios were very similar, although road deposition was only modelled in the second case.
Therefore, deposition rates within fields, patch-borders, colluviums, and valley-floors for the connector
scenario were ~30% higher than for the other simulations. As the sediments not diverged by the road
network were ultimately deposited within the catchment, the sink and connector scenarios displayed
very similar hillslope sediment yields. Contrarily, sediment yields for the shortcut scenario were in
general ~4.5 times higher than for the remaining road connectivity simulations.
Table 3. Summary statistics of soil redistribution rates, hillslope sediment yields calculated by the
WaTEM/SEDEM simulations.

| Scenario | | Erosion | | | Deposition | | | SSY | | | SY | | |
|---|---|---|---|---|---|---|---|---|---|---|---|---|---|
| | | -------------------- Mg ha$^{-1}$ yr$^{-1}$ -------------------- | | | | | | | | | --------- Mg yr$^{-1}$ --------- | | |
| | | Q1 | Q2 | Q3 | Q1 | Q2 | Q3 | Q1 | Q2 | Q3 | Q1 | Q2 | Q3 |
| Connector | GS | 3.5 | 6.3 | 8.7 | 3.4 | 5.9 | 8.3 | 0.2 | 0.3 | 0.5 | 1,047 | 2,248 | 3,307 |
| | NGS | 3.7 | 6.6 | 9.1 | 3.5 | 6.1 | 8.5 | 0.2 | 0.4 | 0.6 | 1,498 | 3,054 | 4,097 |
| Shortcut | GS | 3.5 | 6.3 | 8.8 | 2.7 | 4.9 | 7.2 | 0.6 | 1.2 | 1.8 | 3,878 | 8,467 | 12,242 |
| | NGS | 3.7 | 6.6 | 9.2 | 2.5 | 4.7 | 6.7 | 0.9 | 1.9 | 2.6 | 6,303 | 13,238 | 17,506 |



| | | | | | | | | | | | | |
|---|---|---|---|---|---|---|---|---|---|---|---|---|
| Sink | GS | 3.5 | 6.3 | 8.8 | 3.4 | 6.0 | 8.4 | 0.1 | 0.3 | 0.4 | 833 | 1,828 | 2,665 |
| | NGS | 3.7 | 6.6 | 9.2 | 3.5 | 6.2 | 8.7 | 0.2 | 0.4 | 0.5 | 1,143 | 2,389 | 3,197 |

SSY: area-specific hillslope sediment yield; SY: hillslope sediment yield. Deposition rates include
hillslope and road deposition. GS: grass-buffer-strips; NGS: no grass-buffer-strips; Q1: first quartile, or
the 25th percentile; Q2: second quartile, or the median; Q3: third quartile, or the 75th percentile.
The comparison between WaTEM/SEDEM simulations and the average annual loads from the
monitored tributaries revealed a larger overlap between latter and the results from the 'road-as-shortcuts'
scenario (Figure 7). For this comparison, we only considered the simulations with the presence of grass-
buffer-strips, which more closely represent the actual structure of the agricultural fields in the Baldegg
catchment (see Figure 2). The overlap became particularly clear then we compared the interquartile
range (IQR) of the calculations (Figure 7). That is, only a small proportion of the 'road-as-connectors'
and the 'road-as-sinks' model realisations encompassed the IQR of the tributary sediment loads, except
for the Höhibach, which showed the opposite pattern.

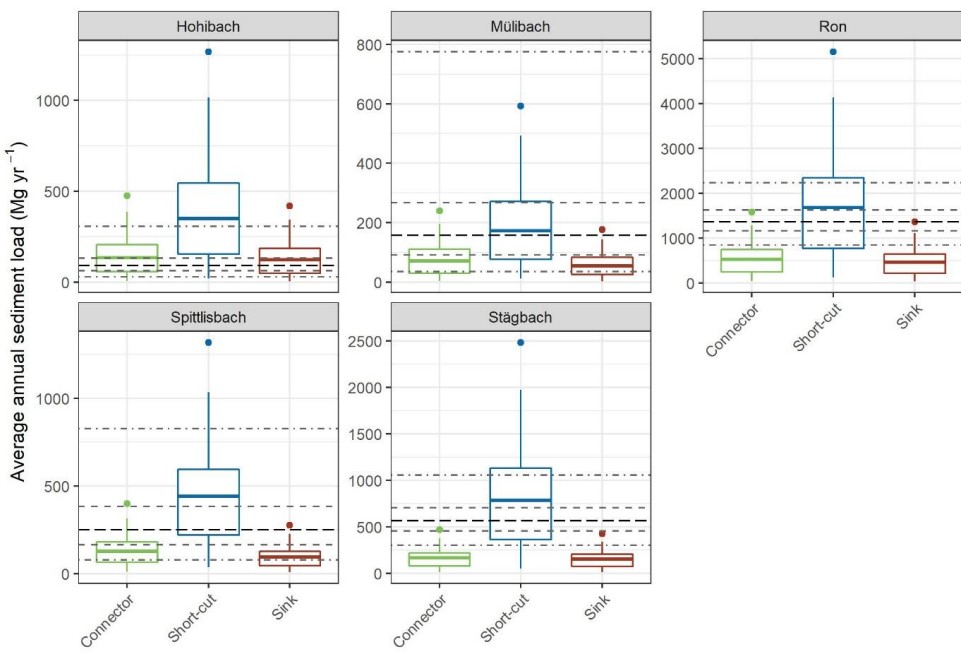


Figure 7. Box-plots of hillslope sediment loads simulated by WaTEM/SEDEM for the road connectivity
scenarios for each tributary sub-catchment. Dashed lines represent the percentiles of the sediment loads
for each tributary, calculated based on the error propagation of the sediment-rating curve.
It is important to note that the median sediment concentrations calculated by the rating curve (Equation
1) underestimated the actual observations, for all tributaries. This is expressed by the positive mean error





of the estimates (Table 4). Moreover, the Nash-Sutcliffe model efficiency coefficient for the median
calculations was unsatisfactory considering the usual thresholds for model performance (e.g. Moriasi et
al., 2015). On the other hand, the 95 % prediction interval of the rating curve encompassed a large
proportion of the observations, and most errors were associated to extreme events (Table 4, Figure 8).
Hence, it is likely that actual sediment loads from the tributaries are contained within the long right side
of the skewed distributions resulting from the error propagation of the rating curves (Figure 8).
Table 4. Evaluation metrics of the sediment rating curve, considering the measured sediment
concentrations and median of the simulations.

| Stream | ME | RSME | Out of bound percentage* | $r_p$ | $r_s$ | NSE |
|---|---|---|---|---|---|---|
| | ----- mg L$^{-1}$----- | | --------- % --------- | ---------------------------------------- | | |
| Höhibach | 50.10 | 80.60 | 0.13 | 0.52 | 0.64 | 0.22 |
| Mülibach | 72.97 | 138.32 | 0.11 | 0.64 | 0.73 | 0.34 |
| Ron | 22.00 | 54.61 | 0.61 | 0.63 | 0.77 | 0.38 |
| Spittlisbach | 95.67 | 149.78 | 0.22 | 0.51 | 0.67 | 0.20 |
| Stägbach | 25.05 | 67.14 | 0.36 | 0.50 | 0.70 | 0.19 |

*percentage of observations out of the 95 % prediction interval. ME: mean error; RMSE: root-mean-
square error, $r_p$: Pearson's correlation coefficient, $r_s$: Spearman's correlation coefficient; NSE: Nash-
Sutcliffe model efficiency coefficient.



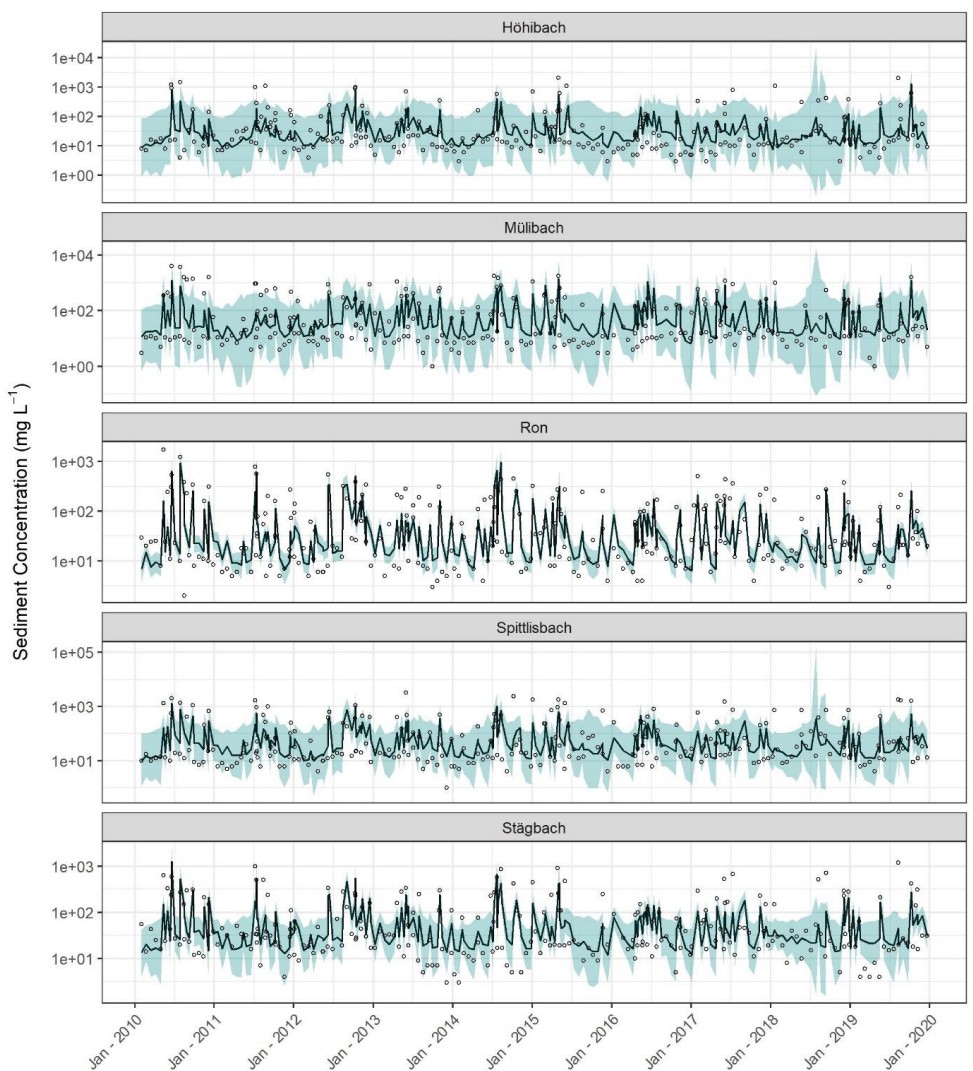


Figure 8. Log-scaled daily sediment concentrations estimates from the rating curve: dark solid line is the median of the calculations and the shaded light blue represents the 95 % prediction interval. Open circles are the observed values used for fitting the curve.

## 4 DISCUSSION

Here we assessed the interaction between landscape patchiness, linear structures, and sediment connectivity. Our quantitative model-based approach highlighted the importance of roads in (dis)connecting sediment fluxes between landscape compartments and surface waters in patchy agricultural catchments, which are typical of the Swiss Plateau. These findings are very much in lines with long-term field observations and qualitative model assessments for similar areas in Switzerland.



For instance, Ledermann et al. (2010) monitored off-site erosion in multiple fields from different regions
of the Swiss midlands, and found that linear features in general and roads in particular had a large
influence on runoff concentration, soil erosion rates, and off-site damage. These authors also estimated
that > 50 % of eroded soil was deposited in adjacent fields and infra-structure, while up to 20 % reached
surface waters, mainly through indirect inflow via the road and drainage network. Such figures are
proportionate to WaTEM/SEDEM estimations for the Baldegg catchment, specifically for the shortcut
scenario with the presence of grass-buffer-strips (Table 3). Another interesting similarity between our
outputs and the field assessments from Ledermann et al. (2010), was that both approaches identified
field border structures as critical regulators of soil erosion and sediment transport (see Figures 5 and 6).
According to the field assessments, border furrows are specifically important for both triggering erosion
and promoting diffuse sediment deposition. Such features, combined with long-term tillage erosion,
might be responsible creating the topographic pattern displayed in Figure 6d.
Moreover, the capacity of roads to connect runoff and sediments from arable land to surface waters in
Switzerland was extensively described by Alder et al. (2015) and Schönenberger and Stamm (2020).
Both studies used a similar semi-qualitative modelling approach for identifying agricultural fields that
were directly or indirectly (i.e. via the road and drainage networks) connected to surface waters. In
particular, Schönenberger and Stamm (2020) mapped the location of drainage inlets in multiple small
catchments of the Swiss Plateau. Accordingly, these authors identified the road drainage system as the
main hydrological shortcut connecting fields to water courses, as most drainage inlets discharge into
surface waters (87%), and only a small proportion of them flow into wastewater treatment plants or
depositional areas. Hence, the fact that the WaTEM/SEDEM 'road as shortcuts' scenario displayed a
greater agreement with the sediment rating curves for the Baldegg tributaries (Figure 7) is coherent with
the current understanding of runoff dynamics in the Swiss Plateau. Of note, the contrasting results for
the Höhibach sediment loads (Figure 7), which are much closer to the sink and patch-connector
simulations, do not seem to be explained by any physiographical specificity of the sub-catchment.
Hence, we speculate that this different pattern could be caused by a lower inlet drainage density in the
Höhibach sub-catchment, or by in-stream process, that are not accounted for in WaTEM/SEDEM.
In addition, our simulations of edge-of-field grass-buffer-strips indicated that these structures might be
particularly relevant for the 'road as shortcuts' scenario. In this case, the model estimated that grass-
trips could reduce up to 30% the sediment connectivity from hillslopes to surface waters in the Baldegg
catchment (Table 4). However, it should be noted that we assumed 2 m wide strips at field-block borders,
irrespectively of the adjacent structures or land use. As previously mentioned, the extent of these features
is in fact quite variable, and legislation only requires 0.5 m filters between fields and roads, as reported
by Alder et al. (2015). These authors further emphasised that albeit edge-of-field strips are an important
complementary management practice, their effectiveness is often reduced at high inflow areas, in which
very wide buffers would be necessary to stop sediment fluxes. Hence, Alder et al. (2015) recommended



that minimising on-site erosion rates was ultimately the most effective way to decrease sediment input
from arable land to water courses in Switzerland. Our results support this management proposition.
However, our simulations also indicate that the disproportional sediment connectivity afforded by the
dense road network translates into an excessive sediment supply to water courses, even when simulated
erosion rates were small. As on-site erosion rates in Switzerland are already reasonably low (see
Prasuhn, 2020), it might be important to consider solutions that address the sediment transport through
the underground drainage system, particularly in environmentally sensitive areas, such as the Baldegg
catchment.
In a wider context, our study has demonstrated how structural sediment connectivity patterns can be
investigated with a conceptual model as WaTEM/SEDEM, provided that model resolution is sufficiently
fine to represent relevant features and processes. In the Baldegg catchment, and likely in other patchy
agricultural landscapes, soil redistribution rates and patterns are intrinsically linked to linear features.
Hence, in order to provide relevant system descriptions, soil erosion models applied under similar
conditions must be able to represent linear features and landscape patchiness. Although these results
might seem case-specific, similar findings have been reported around the world. For instance, the effects
of roads and farm tracks in both coupling and decoupling runoff and sediments has been described in
Australia (Croke et al., 2005), Brazil (Bispo et al., 2020), Kenya (Stenfert Kroese et al., 2020), Italy
(Persichillo et al., 2018), and Spain (Calsamiglia et al., 2018). Moreover, the influence of linear features
such as field borders, hedges, terraces, and tractor tram lines in soil redistribution rates and patterns have
been well-documented in Europe (Calsamiglia et al., 2018b; Evrard et al., 2009; Fiener and Auerswald,
2005; Lacoste et al., 2014; Saggau et al., 2019), as well as the importance of landscape structure
(Baartman et al., 2020; Chartin et al., 2013; Fiener et al., 2011).
Another generalisable finding from our research was that WaTEM/SEDEM can be as sensitive to
RUSLE parameters as to the model-specific transport capacity coefficients. Therefore, when performing
uncertainty analyses of WaTEM/SEDEM, it is important to consider sources of error associated to the
RUSLE parameterisation. So far, uncertainty estimation methods applied to WaTEM/SEDEM have
focused on the $K_{TC}$ parameterisation, and therefore have underestimated the uncertainty in model
predictions. We anticipate that our open-source WaTEM/SEDEM script will facilitate stochastic
implementations of the model, and ultimately promote uncertainty and sensitivity analysis of soil erosion
models. As recent studies have again demonstrated, results from soil erosion models are only
interpretable if the uncertainty in model structures, parameter estimation, and observational forcing data
are accounted for (Eekhout et al., 2021; Schürz et al., 2020).
**5 CONCLUSIONS**
Here we employed a global sensitivity analysis of the WaTEM/SEDEM model to investigate the
influence of linear structures and landscape patchiness on sediment connectivity in the Baldegg
catchment, a representative area of Swiss Plateau. In particular, this novel application of


WaTEM/SEDEM was implemented with the free programing language R, and our code is available as
supplementary material.
Our results demonstrated that assumptions about road connectivity were by far the most important factor
for modelling sediment transfer in the Baldegg catchment. Moreover, the comparison between extensive
model simulations and sediment rating-curve calculations indicated that roads behave as conduits for
sediment transport in the catchment. Hence, representing road connectivity is crucial for modelling
sediment transfer from hillslope to water courses in this agricultural catchment of the Swiss Plateau, and
potentially in other areas with a dense road drainage system. Moreover, our results further highlighted
the effects of linear structures and landscape patchiness on sediment connectivity. These findings were
made possible by the use of a model that was specifically tailored to explore the particularities of our
study area, by effectively exploring model assumptions and the parameter space, and by the use of high
resolution spatial data.
Overall, we found that WaTEM/SEDEM was useful for investigating sediment connectivity in the
Baldegg catchment, as it allowed us to unravel some of the processes and structures regulating hillslope
sediment transport in the area. If these processes and structures are accounted for, the model shows
potential for upscaling. In the case the model is be used for prediction and decision-making, we
recommend employing a fit-for-purpose rejectionist model testing framework, with multiple sources of
data, in order to evaluate the model's numerical accuracy and the quality of its spatial predictions.
**6 CODE AVAILABILITY**
The code for the model simulations was uploaded as a supplementary material file. If the manuscript is
accepted, we will upload the R script file and input data used for the simulations to the EnviDat platform
(https://www.envidat.ch).
**7 DATA AVAILABITLY**
If the manuscript is accepted, we will upload the input data used for the simulations to the EnviDat
platform (https://www.envidat.ch). This includes:
-    Processed DEM
-    Edited land cover rasters with the locations of grass buffer strips
-    Road network map
-    Field block map
The raw water discharge and sediment concentration data is property of the Department of Environment
and Energy of Canton Luzern, and can be shared upon their discretion.
**8 AUTHOR CONTRIBUTIONS**



PVGB and PF developed the model code, PVGB performed the simulations and analysed the data. SS prepared model input data. PVGB prepared the manuscript with contributions from all authors. CA revised the manuscript.

## 9 COMPETING INTERESTS

The authors declare no conflict of interest.

## 10 ACKNOWLEDGEMENTS

The authors would like to thank Robert Lovas, from the department of environment and energy of Canton Luzern, who supplied the sediment concentration and water discharge monitoring data used in this manuscript. We also appreciate the help from Axel Birkholz in acquiring the data. PVGB would like to thank Franz Conen and Claudia Mignani for their multiple and valuable inputs regarding the conceptualisation and preparation of this manuscript.





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
