# Peer review of "A conceptual model-based sediment connectivity assessment for patchy agricultural catchments"

_Hydrology and Earth System Sciences, 2021_

## Author Comment (AC1)

For clarity and improved visualisation, the reviewer comments are shown from here on in black. The authors' replies are in blue font below each of the reviewers' statements. The changes in the revised manuscript are displayed in green.

**Reviewer #1**

**General Comments**

The paper is excellent in terms of organization, writing, and content. The results and figures are convincing, and I am particularly excited about how the authors utilize changes in model structure to address sediment connectivity, which is timely and quite important in terms of advancing watershed sediment simulations. Overall, I have very minor comments regarding some clarifications in a few instances and I believe that a few statements made by the authors should be relaxed a bit. Additionally, I would suggest including a brief paragraph at the end of the discussion regarding limitations of the WaTEM/SEDEM approach, and how we can further move to improve the spatial and temporal resolution of sediment connectivity simulations.

Many thanks for the time and effort put into reviewing our manuscript. We are pleased to know it was appreciated. Below we respond to all your specific comments. We will also include a paragraph in the discussion further highlighting the limitations of the WaTEM/SEDEM approach and some thoughts on how to improve sediment connectivity simulations.

**Specific Comments:**

L 32: Perhaps you can also mention that the paper is, for the first time to my knowledge, advancing tools to assess connectivity by quantifying structural uncertainty within the sediment simulations (not referring to *structural* connectivity here, by the way, just how there are inherent uncertainties within how the model is configured to predict fluxes/loads).

Thanks for pointing this out. We will highlight how the quantification of structural model uncertainty is a novel aspect of the paper.

L 159-167: It would be helpful within the text to tell readers the temporal resolution of the model. It seems like it's yearly according to the RUSLE equations but could be clarified.

Apologies for this omission. Indeed, the model is operating in a yearly time-step. This will be clarified in the revised manuscript.

L 180: Is this of the individual pixel or along the slope length?

The slope gradient and the transport capacity are calculated per individual pixel.

L 188: Does this include bank erosion?

We will include an explicit mention to bank erosion here, as this process is also not simulated by the model.

L 188-190: If this is a yearly model then perhaps this statement can be slightly relaxed... For example if the system is known to not aggrade or degrade over longer-term (decadal) timescales then instream erosion and deposition are approximately in equilibrium and so I would not be as concerned with the instream.

We completely agree with your comment and that our statement could be somewhat relaxed. We would like to rephrase to:

"Since WaTEM/SEDEM does not represent gully and bank erosion or in-stream erosion and deposition processes, any comparison between modelled sediment yields and catchment-outlet sediment loads must be interpreted with caution. However, in catchments where rill and interrill can be assumed to be the main erosion processes, and assuming a state of fluvial quasi-equilibrium due to the temporal resolution of the analysis and the model, the outlet sediment loads should be at least comparable to the model outputs, even if not fully commensurable".

L 205: How did you decide 1200? Is this enough? Sometimes people will utilize 100,000 monte carlo simulations. I'm not saying that you need to run the model for more realizations, just a bit more justification please.

In all honesty we do not have a strict justification for the number of iterations in the Monte Carlo simulation. We understand 1200 iterations was enough to explore the parameter space for the purpose of the sensitivity analysis. If we wanted to sample the parameter space exhaustively in a rejectionist approach, then we would likely need a higher number of simulations considering the number of parameters.

L 205-208: Right, the typical approach is to calibrate the model and along the way assess sensitivity/uncertainty such that sensitivity/uncertainty of the model is addressed within solution spaces that are plausibly behavioral. I'm not rejecting your approach by any means,

but perhaps some additional acknowledgement of the traditional approach and how you are slightly deviating here could be helpful to readers. Some readers might question why you present realizations that will not adequately describe the sediment load/flux in the system. L 208: *model assumptions* – I would clarify you are making assumptions about the structure of the model, so quantifying structural uncertainty.

Precisely – with the sensitivity analysis we did not aim to identify the behavioural parameter space, but rather to understand how the model responds to the different structural assumptions. We will make the following changes in this paragraph:

"Our model application consists of a global all-at-a-time sensitivity analysis, as described by Pianosi et al. (2016). That is, we performed a Monte Carlo simulation to explore the variability of the whole parameter space, and all input factors were sampled simultaneously for each model realisation (n = 1200). The framework is similar to an uncertainty analysis, except in this case we did not focus on locating the parameter space which produced behavioural model realisations. Instead, we concentrated on apportioning sources of uncertainty to different model input factors, aiming to rank their contribution to the variability of the response surface (see Pianosi et al., 2016 for a review on sensitivity analysis). This should allow us to identify parameters and model assumptions that have a greater impact on the manner with which WaTEM/SEDEM describes sediment connectivity in the Baldegg catchment. In particular, the analysis of different assumptions about the structure of the model should provide a connectivity assessment based on the apportionment of the structural uncertainty withing the simulations. To the best of our knowledge, this is the first time model structural error is incorporated into sediment connectivity research."

L 238: can you clarify why a value for *Pcon* wouldn't be applied everywhere in the catchment, but instead for just the forest and buffer strips? What if there is disconnectivity from microtopography in the roadside ditches, for example? Again – I'm not asking for additional analyses, just a sentence or two for clarification and that you *might* parameterize this other places in the watershed if you had overt reason to.

This is a great point, thanks for bringing it to our attention. The parcel connectivity parameter was originally developed to represent the extent with which water and sediment transport is reduced at parcel borders in case the downslope patch is composed of forests or grasslands. We will include this clarification in the text. We completely agree that the parameter could be

incorporated in other places, and we will mention this in the model limitations/improvements section you suggested.

L 255-259: can you please add a sentence that details the difference between scenario two and three? The way I understand it is that in scenario three sediment deposition does not occur on the road or in swales/ditches along side the road, but deposition can still occur downstream, for example in between the road and the stream network. In scenario two sediments are automatically connected to the stream, correct?

That is absolutely correct, we will include this clarification in the text:

"For this scenario, deposition will never occur on road cells, however sediments can still be deposited on downstream patches, before reaching the stream network".

L 270: Again I might suggest using the word *structural* uncertainty of the model.

Thanks, we will include a mention to structural uncertainty here:

L 275-276: This was a bit confusing to me. L 306: What is the mean squared error in relation to? The yearly predicted sediment load and the yearly average sediment load from the rating curves? Please clarify.

Apologies for this confusion. The random forest analysis was used to predict the WaTEM/SEDEM simulations of hillslope sediment yield, based on the parameter values that were sampled for each iteration. The mean squared error is calculated from the RFA predictions and the WaTEM/SEDEM simulations. The increase in error due to the absence of a variable is used to rank its importance. We will clarify these points in the methods and the results.

L 323: Fig. 6g – what about interrill erosion?

Thanks for noticing this, we change it to rill/interrill erosion.

L 323: Fig 6b,c; L 333: Is it worth showing land use for all the details here?¨

Yes, that is a good idea, thanks. We will add an identification on the landuse in figures 6b,c.

L 374: Perhaps you can say in the caption that the short-cut generally overlaps the IQR better than the other 2 scenarios... this could help readers quickly interpret the figure.

Thanks, we will add that to the caption.

L 385: Out of bound percentage – is this a fraction or a percentage what is presented in the table?

Apologies for this mistake, indeed we were presenting the fraction. This will be corrected accordingly.

L 388: It would be great if we could see at this same time scale how SEDEM was performing… but I think this is just a limitation of the model since it runs at a yearly scale, correct?

Yes, exactly. For this reason, we only made comparisons with the average yearly loads.

L 424: Perhaps also the rating curve is underestimating the load, as you previously mentioned? Which would improve the performance of simulations with respect to the short cutting, correct?

We are not sure this would be the case here, as the curves are probably underestimating the actual loads for all streams – not just for the Hohibach.

L 451: I believe Mahoney et al., 2018 talks about importance of road networks a bit in the USA.

Thanks, we will include this reference here.

L 463-465: I think this last sentence should be relaxed a bit... quantifying all of the sources of uncertainty due to observation data, model input data, model output data, parameter uncertainty, etc. etc. is quite the undertaking. In fact, in my opinion, it might be an impossible task. Does this invalidate the use of models, however? In my opinion, no, it does not. We can still discern important information from models even though we don't account for 100% of uncertainties. It ultimately will depend on what questions we are trying to answer with the model and what the model is attempting to do, which can be equally as important as quantifying certain uncertainties in my opinion.

We completely agree that accounting for all uncertainty is simply impossible. What we were trying to convey here is that soil erosion and numerical connectivity models are highly uncertain. This uncertainty stems from multiple unknows about the modelled phenomenon, the input data, and the forcing data. Our opinion is that neglecting such uncertainty makes it very

difficult to provide meaningful insight based on the modelling. In any case, we see how our statement might have been too strict here, and we will rephrase considering your comment.

L 465-466: It would be nice if a paragraph on limitations of the modeling approach and future opportunities could be included. For example, while RUSLE is relatively easily implemented and approachable, it would be nice if the RUSLE approach was a bit more physically based. Additionally the RUSLE approach limits the temporal resolution of the model, so seeing event- and seasonal-scale connectivity seems a bit limited. Furthermore, the advanced geospatial data that facilitates this novel connectivity modeling is wonderful, and can help to elucidate hotspots of connectivity. Additionally there is recent sentiment to move towards high-temporal resolution models to quantify hot-moments of connectivity. The yearly timescale inherent to the RUSLE approach perhaps is underserving this sentiment.

We completely agree. We think a model like WaTEM can be useful to explore structural connectivity patterns, but much more dynamic models are needed to quantify these hot-moments of connectivity, and to get a better grasp of the functional connectivity of the system. As we previously stated, we will include the paragraph you suggested.

L 466: I'd suggest perhaps emphasizing that exploring structural uncertainties in the model framework - and not just parameter uncertainties, as is the traditional method - allowed for advanced understanding of connectivity processes. This type of approach in my opinion is quite underserved in modeling work and should be considered in the future where high-resolution geospatial data is available.

Thanks for pointing this out throughout the manuscript. We will emphasize here and in the remainder of the manuscript the relevance of quantifying structural uncertainties.

Technical Comments:

L 312: I'm not sure if the different colors are helpful here, maybe consider symbols?

The colours are just there to keep the theme of the graphs – the facet titles identify the scenarios. If its ok, we would like to keep them.

L 324: typo

Thanks, corrected.

L 364: typo, confusing

We will rephrase to: "The comparison between WaTEM/SEDEM simulations and the sediment loads revealed a larger overlap between the latter and the results from the 'road-as-shortcuts' scenario (Figure 7)."

---

## Author Comment (AC2)

For clarity and improved visualisation, the reviewer comments are shown from here on in black. The authors' replies are in blue font below each of the reviewers' statements. The changes in the revised manuscript are displayed in green.

**Reviewer #2**

**General comments**

This paper provides a modelling approach to analyse the effect of linear features on sediment connectivity in a mesoscale catchment. The authors put a lot of work into adapting an existing model such that it is able to account for these effects. I appreciate this effort and I think that this work is important for improving the understanding of sediment connectivity in agricultural catchments. However, I have some major points of criticism which are the following ones:

We highly appreciate the time and effort put into reviewing our manuscript. We specifically appreciate the attention to detail and the discussion about what conclusions can be drawn from the study. We address all your comments below.

The authors state themselves that a comparison between their model results and the measuremed sediment loads in the river should be performed with upmost caution. However, in the results and in the discussion they still make strong conclusions based on exactly such a comparison. The authors should therefore reformulate the discussion such that it reflects this uncertainty better.

We apologise if this were conveyed: we tried to be very cautious about the conclusions drawn simply from the comparisons with the sediment loads. Overall, the importance of linear features for regulating sediment connectivity has been extensively documented by field observations and modelling approaches in Central Switzerland (Alder et al., 2015; Ledermann et al., 2010; Remund et al., 2021). Results from our study corroborate these observations, both due to the sensitivity analysis and the comparison with the sediment loads, which we understand were cautious (or even overly cautions considering some of the comments from reviewer #1). As we state in the manuscript, the comparison between outlet sediment loads model outputs provide a general picture of the plausibility of the simulations.

In any case, we will reformulate the sentences you mentioned in the specific comments in order to be more careful about any potentially overreaching conclusions. This is explained below in our reply to your specific comments.

The comparison between the model results for the different scenarios and the measured sediment loads does not make sense for me in the way it is done currently. In their current analysis, the authors ignored important factors influencing the sediment connectivity in the catchment: Firstly, as I understand it from the manuscript, they treated all grassland areas like arable land areas in the model. Since around a third of the catchment area is covered by grassland, this leads to a large overestimation of sediment loads.

Here we believe there has been a misunderstanding, which is entirely our fault. Indeed, we did not separate grasslands from croplands (reasons why are explained in the specific replies). However, we considered this uncertainty in the parameterisation of the CP factor, which had the lower limits of their "prior" distribution stretched to include typical values for permanent grasslands in Switzerland. We apologise, as this could not be understood from the methods. Similarly, the Ktc high parameter was sampled from 0 to 200 m to cover a wide parameter space. Hence, some model realisations will indeed overestimate the erosion rates, whilst others might underestimate them. Since we did not focus on calculating exact erosion rates for the catchment, but rather on assessing the influence of linear features and landscape patchiness in sediment connectivity, we do not see any major limitations with our approach. Particularly since all this uncertainty is incorporated into the model results.

Secondly, the authors assumed a two-meter grass buffer strip around all agricultural plots. The authors state that they don't know the real width of buffer strips in the field and that they therefore use a value of two meters for testing the sensitivity of the model. Even though the sensitivity analysis showed that the buffer width had a large influence on the model results, the comparison between modelled and measured sediment loads is only done for the two-meter scenario. Also the conclusions are only drawn based on this scenario. Even though the authors state that the two-meter scenario is more realistic than the scenario without a buffer, this value contradicts the values reported by other studies (e.g. Alder, 2015; Remund, 2021) and the legal requirements. Although this point is addressed in the discussion, I am missing a proper justification why the measurements are only compared to the two-meter scenario.

We assumed a 2 m width to test the sensitivity of the model to the presence of buffer strips. The value was pragmatically chosen based on the spatial resolution of the model input data – apologies for not mentioning this in the methods. In addition, we understood it was better having one fixed value for testing such sensitivity, than having multiple values which would anyway be uncertain. To clarify, our sensitivity analysis did not demonstrate the model was sensitive do the buffer *width*, as we only tested one value. We think testing additional values goes beyond of the scope of our research.

Now, if the buffer strips alongside roads in the catchment (which are at least 0.5 m wide according to legislation, but highly variable in width) are completely ineffective, then the scenario without grass strips would possibly be the most appropriate representation of the system – at least considering the capacity of model outputs to mimic the data. However, the strips would still be there in the fields, and this was our rationale when we stated that the scenario with strips "more closely represent[s] the actual structure of the agricultural fields in the Baldegg catchment". As all results are summarised in Table 3 of the manuscript, and the scenario without strips shows the exact same pattern for the different road connectivity assumptions as the scenario with the strips – leading to the same conclusions – we thought it would interesting, for concision and clarity, to focus the discussion on the latter.

Accordingly, we propose to add a similar graph as in Figure 7 for the scenario without strips in the supplementary material, and to explain this choice in the revised manuscript.

Thirdly, in their "shortcut" scenario, the authors assume that all roads and farm tracks are drained with shortcuts. (At least, this is how I understand it from the manuscript.) I expect that a major part of the roads in the Baldegg catchment is actually not drained by shortcuts. Therefore, it makes sense to use the current "shortcut" scenario in a sensitivity analysis, but not as a realistic scenario. Compared to reality, the current "shortcut" scenario is expected overestimate the real sediment loads. Consequently, even though the "shortcut" scenario is most similar to the measurements, this is possibly simply caused by a bias in the model input. Even though I agree that roads and shortcuts are in fact important for sediment transport, I think this cannot be concluded from the current analysis. To state that "roads behave as conduits for sediment transport in the catchment", as it is done in L474f, it is inevitable that the scenarios are revised such that they reflect the reality in the catchment better. (At least for the first point mentioned. Second and third point may also be discussed.)

We completely agree that in reality only a certain portion of the roads will drain the sediments directly to the stream network (note that we never used the word "realistic" to describe any of the scenarios). However, as we explained above, we used quite a wide range of parameter values and different scenarios to examine *how things could happen* in the catchment. Still, the model was only able to provide reasonably comparable results to the outlet sediment loads if we assumed that roads behave as conduits for sediment transport. And this is considering the large uncertainties in both the sediment rating curves and the model outputs. These findings, in combination with the results from the sensitivity analysis, and the multiple studies that report similar patterns for other sites in Switzerland (as reviewed in the discussion), should allow us to state the following in the conclusion:

"Our results demonstrated that assumptions about road connectivity were by far the most important factor for modelling sediment transfer in the Baldegg catchment. Moreover, the comparison between extensive model simulations and sediment rating-curve calculations indicated that roads and hydraulic shortcuts are likely to behave as conduits for sediment transport in the catchment. Hence, representing road connectivity is crucial for modelling sediment transfer from hillslope to water courses in this agricultural catchment of the Swiss Plateau, and possibly in other areas with a dense road drainage system. Moreover, our results further highlighted the potential effects of linear structures and landscape patchiness on sediment connectivity."

The authors state that the catchment is representative for the Swiss plateau. However, they do not further elaborate on this. Other studies, however, rather suggest that the catchment has a low shortcut connectivity compared to other catchments in the Swiss plateau (see comment to L98). The authors should improve on putting their analysis in the right context.

Thanks for pointing this out. We will remove any mentions in the manuscript about the catchment being representative for the Swiss Plateau. Indeed, that cannot be affirmed, and it is not our objective to provide a representative case study for this part of Switzerland.

Several specific comments (see below) should be addressed to improve clarity and reproducibility of the study. Additionally, the manuscript should also receive some revisions regarding language and correct spelling.

From our understanding, the comments on reproducibility are mostly related to software and package versions, and to methodological details regarding data preparation. These points will be addressed accordingly, as we explain in the replies to your specific comments below. In addition, we have corrected the typos and spelling errors you highlighted. We would be glad to send the manuscript over to a native speaker for review if the editor deems necessary.

**Specific comments**

L36: Talking about a "continuous displacement of small amounts" is wrong here. The discplacement varies strongly between events and years, as you also state below.

We will rephrase to: "Rainfall events on sloped surfaces continuously displace soil particles, which are transported downslope as sediments".

L47: Rephrase.

We would appreciate more guidance here about what needs rephrasing and why.

L56: You should also add the most recent publications here, e.g.: https://doi.org/10.1016/j.catena.2021.105290

Thanks, we think the reference is appropriate and we will include it in the manuscript.

L66: "assuming they are able to explicitly take connectivity into account": Difficult to understand. Please write this more clearly.

Apologies if that was not clear. We will rephrase to: "These usually rely on high-resolution process-based models, assuming they are able to represent connectivity dynamics".

L73: with a size of few square kilometres

Thanks, this will be updated as suggested.

L79: You state above that one major issue of erosion models is the uncertainty of input data. Then you state that you used a high resolution dataset (2x2m DEM). However, for whole Switzerland, a 0.5x0.5m DEM is freely available in the same quality as the 2x2m DEM. You still used the 2x2m DEM. Why did you not use the higher resolution model?

In such high resolutions (0.5 m), the influence of the microtopography becomes much more prominent, and we understood this conceptual model would not be the most appropriate for handling such features. We would like to highlight however that the 2 m resolution we are working with is much higher than what is usually employed in erosion modelling research, in particular at catchment/mesoscale (see Borrelli et al., 2021).

L98: Here you state that the Baldegg catchment is *patch*y and *representative* for the Swiss Plateau. Below, you only elaborate on the patchiness, but not at all on the representativeness. Either elaborate on the representativeness below or use another word here. Schönenberger et al. investigated two catchments in proximity or even inside your catchment. Compared to the distribution in the Swiss plateau, these catchments however rather seem to have a low shortcut connectivity. This indicates in my opinion that also your catchment is rather on the lower side with respect to shortcut connectivity.

We agree that we did not elaborate on the representativeness of the catchment and ultimately this is not our goal here. Hence, we will remove any reference to the catchment being representative for the Swiss Plateau.

L103: In Figure 1c, you use the term "infrastructure", here you use the term settlements. It is unclear how these two terms differ and what the term infrastructure means. Please use consistent terms here. Additionally, how did you treat the areas of roads? Did you include them into the settlement area? Or are they included in the agricultural land/forest area?

Apologies for this inconsistency. Roads and settlements areas were calculated as part of the infrastructure. This will be adapted in the text.

L117: Are tile drainage only located in water accumulation zones? (What are water accumulation zones? Are the determined based on topographic index, slope?)

By definition tile drainage is found where there is excess water. Upon reflection we found this information superfluous, and we will remove it from the revised manuscript.

L120: How did you determine these slopes? Which elevation model? The maximal slopes are strongly depending on the model used. You are referring to Figure 1b. However, the slope is not visible in Figure 1b.

Figure 1b is referred for altitude "higher altitudes are found in the eastern and western sides of the catchment (Figure 1b)" in line 120. Slope was calculated with the same DEM used for the model, which will be mentioned in the revised manuscript.

L122: "in this case formed by the retreat of the Reuss Glacier in the south to north direction (~18,000 years BP)" -> Not important. Consider removing.

We think this information contributes to the description of the study site and would appreciate if we could keep it.

L127: MeteoSwiss -> Please add reference.

We will double-check how to reference the data from MeteoSwiss here.

L132: What is approximately? Provide the range of numbers of grab samples taken.

Apologies, we meant that on average 275 grab samples by stream. We will include more information on the grab samples.

L133: What is "opportunistical sampling"?

This will be removed.

L136: Why 2020? Above you write that you only sampled till 2019.

Apologies, we meant until the end of 2019.

L144: What is k? I guess the covariate ID. This should be written explicitly. Additionally, in contrast to Vigiak & Bende-Michl, you are only using the first five covariates, but not the long-term trend covariates (6 and 7). Why?

Thanks for noticing this. We will explicitly mention that k is the covariate identity. We did not use the long-term trend covariates because they were either not significant or did not improve the models.

L149: First column of table: Remove the word "is". Also the word "water" is not really needed. Second column of the table: This is not the quadratic term of $Q_i$, but of $x_{2,i}$.

Thanks again for noticing these errors. We will correct them as suggested.

L156: You are addressing the variance in sediment concentrations extensively. However, you are not addressing the uncertainty in daily discharge at all. Why?

Essentially because the uncertainty in the sediment load calculations is much larger than for water discharge. We have on average 275 measured sediment concentrations per tributary, which need to be extrapolated for 10 years. On the other hand, daily discharge measurements are available.

L165: Shortly explain why you only focused on water erosion.

We chose to focus on the water erosion instead of tillage because the latter is not relevant for the type of connectivity processes we are investigating. This will be included in the manuscript.

L193: "usually implemented" -> Rephrase. (It is either implemented or not. Possibly, you could write something like that this version is often used.) Provide references where this version is used.

Thanks, this will be rephrased as "WaTEM/SEDEM is implemented as a user-friendly GUI […]".

L199: R version? L200: SAGA version?

Versions will be included in the code, and we will consider using the packages you suggested. We don't think version numbers need to be written in the main text, however.

L201: The code does not contain any information on the versions of the packages used. To make sure that the code can still be used in the future, you should at least provide information on the package versions used. Consider also using tools like packrat, checkpoint or docker. To make the code useful, you should also provide examples of input files.

Thanks for the package recommendations. We will include all information on package versions in the code – apologies for this missing information. As we stated in the data availability section, we will upload the model input data to Envidat if the manuscript is accepted.

L219: It remains unclear how you derived the land cover map. The reference is not shown in the reference list. Therefore, I don't understand if you used a vector dataset that you rasterized yourself or if you used a raster dataset provided by Swisstopo. What does the resolution "1:25:000" mean in the latter case? If you used vector datasets: How did you deal with point and line features, e.g. roads or hedges? Did you assume widths for roads and hedges?

We are terribly sorry for missing Swisstopo in the reference list. Indeed, we rasterised the land cover vector data (Swiss Map Vector 25 BETA, scale 1:25,000) to a 2 m resolution. The roads were firstly converted from lines to polygons with a buffer, considering their width. Hedges and tree lines were already represented as polygons in the vector map. This information will be included in the manuscript.

L220: The statement that spatially distributed crop statistics are unavailable is wrong. There is a plot-resolution crop dataset from the canton of Lucerne available freely for the whole canton (and accordingly for the whole catchment) for the year 2019. Why was this dataset not used? (CP and $K_{TC}$ depend on the crop and you reported them to be the most sensitive model parameters.) Lavrieux et al. state that one third of the agricultural area consists of permanent grassland. Therefore, I expect this decision to have a large influence on your results, leading to an overestimation of erosion. You should address this point at least statistically for each of the five subcatchments analysed and for the full catchment. (e.g. look at the fractions of grassland and reduce the estimated amount of sediment load accordingly). Alternatively, you could also do a spatially explicit analysis.

In all honesty we were not aware of the availability of such data for the canton of Lucerne. We inquired the canton about it, and the geodata would indeed be available. However, this would cost 4260 CHF. For future reference, could you point out where to find this data? We could access the free data using a WMS-link, but we could not perform any spatial analysis with it.

In any case, as we stated before, we apologise for this misunderstanding regarding the grasslands. As we will explain in the revised manuscript, "The minimum CP values were particularly reduced to include typical values for permanent grasslands in Switzerland (~0.01) (Schmidt et al., 2018b), and therefore to represent the uncertainty associated with the land cover classification". This lower limit of the CP parameter is also analogous to the lowest value recommended for permanent grasslands in Europe (0.01-0.08) (Panagos et al., 2015). Similarly, the Ktc high parameter was sampled from 0 to 200 m to sample a wide parameter space. Hence,

some model realisations might overestimate the erosion rates in the catchment, whilst other realisations will likely underestimate them. In either case, what we can see with the model results is that whatever values we sample for the CP or Ktc parameters, the assumptions about connectivity have the highest impact on the model results.

L226: Why did you use the 2x2m resolution DEM? (see comment to L79) How did you process the DEM, e.g. sink filling?

The choice of DEM resolution is explained in the comment to L79. The DEM was sink filled by the Wang & Liu (2006) method, which is implemented in SAGA.

L227: In the $K_{TC}$ column, consider indicating the land use classes belonging to "high" and "low" (e.g. in brackets). This would make it much easier to read.

Thanks, we will include the information in brackets.

L230: Specify that this relates to the maximum CP factor.

We will mention that in the revised manuscript.

L233f: You talk about forests and grass buffer strips in the land cover map. However, the reader is missing were you explain the derivation of forests and grass buffer strips. Consider stating that you are explaining this below.

We are not sure what you mean by derivation of forests and buffer strips, so we would like to maintain this part of the text as it is.

L246: How much wider? Refer to the article in the corresponding legislation directly, instead of Alder et al..

Sorry for the missing information. We will rephrase to: "The extent of the buffer-strips in reality is quite variable, and generally wider at forest and river vicinities (3 – 6 m), as required by law in Switzerland (Alder et al., 2015)."

We would like to keep the reference to Alder et al., which provides this and other information about grass buffer strips in Switzerland, in English.

L249: What were your assumptions on buffer strips along hedges? Did you also use a 2m buffer? Or a buffer corresponding to the legal requirements? How did you treat tree lines?

We used a 2 m buffer along the hedges and treelines, which were rasterised from the land cover map mentioned previously.

L250: As mentioned in the comment to L219, I don't really understand how you derived the road areas.

The road widths are provided in the land cover vector data (Swiss Map Vector 25 BETA). We used these widths to perform a buffer around the road lines. Next the polygons were converted into a 2m resolution raster (same resolution as the DEM).

L252: What is "infrastructure"? Was this also derived from the land cover map?

Infrastructure includes roads and settlements. This will be informed in the text.

L253: If roads act as sinks, why is this related to field drainages? From the text, I don't understand where you assume the sediments to be trapped. On the road? In the drainage system? In sludge collectors? The scenario seems to make sense for me, but you should specify your assumptions more clearly.

We will rephrase to "This represents a scenario in which roadside ditches and the road drainage system trap most sediments and partly diverge runoff to wastewater treatment plants".

L256: Did you assume here that all roads are acting as shortcut? Or only a part of the roads?

We assumed all roads behave as shortcuts.

L262f: How many directions?

Information on the multiple flow direction algorithm implemented in SAGA can be found in Freeman (1991) and Quinn et al. (1991). It allows for divergent flow paths, differently to the typical D8 approach.

Freeman, G.T. (1991): Calculating catchment area with divergent flow based on a regular grid. Computers and Geosciences, 17:413-22.

Quinn, P.F., Beven, K.J., Chevallier, P. & Planchon, O. (1991): The prediction of hillslope flow paths for distributed hydrological modelling using digital terrain models. Hydrological Processes, 5:59-79.

L266-268: In my opinion, some (or probably all) of the additional packages (e.g. "doParallel", "foreach") are not worth mentioning here, as they are only used to speed up the calculation process, but not important for reproducibility of your work.

Thanks, we will remove these references.

L278: What does RFA stand for?

Apologies, RFA stands for random forest analysis. We have updated the text.

L280: Version?

Information on package versions will be included in the code.

L299: Do I understand correctly that Mg yr$^{-1}$ means tons per year? Probably write tons instead.

The megagram (Mg) is equal to a metric ton. It is common approach to express erosion and sediment transport rates with such unit.

L299: For me, the differences between GS and NGS scenarios are not well visible in the plots. Consider making this better visible, e.g. by adding a moving average per category or something similar.

We think a moving average would not the best choice here, as the idea with the scatter plots is also to display the spread of the response surface.

L317-L318: How did you quantify this for the whole catchment? Just by eye? If yes, you should provide the respective plots (e.g. in the appendix). Otherwise, can you provide a quantitative assessment?

Figure 6b, which we refer to, provides an example of the mentioned depositional patterns. In addition, increased within-field deposition rates are expressed quantitively in Table 3.

L324: In Figures 6b, 6c, 6e, 6f, and 6g, arrows indicating the flow direction would help to understand the plot better.

Thanks for the suggestion, we will include arrows indicating the flow direction.

L358: In my opinion, Table 3 would be understandable easier if you would write 25%, 50%, and 75% instead of Q1, Q2, Q3.

Thanks, but in this case, we would like to keep the quartile numbers in the table header.

L366f: As I understand from L244ff, you say that you don't really know what the real widths of buffer strips are in the catchment. It therefore makes sense to me that you use a fixed width of 2m and use it for testing the model sensitivity by running two scenarios – one with and one without the buffer strip. However, in L366 you now state that the 2m scenario is more realistic than the "no buffer" scenario. I agree that a 2m scenario is probably the more realistic scenario along forests. However, I expect the effect of a grass buffer along forests to be small as sediments are trapped by forests anyways. Along roads, I doubt that the 2m scenario is more realistic than the "no buffer" scenario, since the legal requirement is only 0.5m (as you write in L430). (However, I might be wrong with these doubts.) Since I expect the buffer width along roads to be much more important for your model results than the ones along forests, I think you should report both scenarios here, or give a clear explanation on why you think or why you can show that the 2m scenario is more realistic.

Thanks for these considerations. As we stated previously, if the buffer strips alongside roads in the catchment, which are at least 0.5 m wide according to legislation, but highly variable in width, are completely ineffective, then the scenario without grass strips would possibly be the most appropriate representation of the system – at least considering the capacity of the model to mimic the outlet data. However, the strips would still be there, and this was our rationale to state that the scenario with strips "more closely represent[s] the actual structure of the agricultural fields in the Baldegg catchment". As all results are summarised in Table 3 of the manuscript, and the scenario without strips shows the exact same pattern for the different road connectivity assumptions as the scenario with the strips – leading to the same conclusions – we thought it would interesting, for concision and clarity, to focus the discussion on the latter.

Accordingly, we propose to add a similar graph as in Figure 7 for the scenario without strips in the supplementary material, and to explain this choice in the revised manuscript.

L379-382: In the "Ron" stream, the 95% prediction interval seems much narrower than in the other rivers (Figure 8). Therefore, the observed values are mostly outside of this interval and the out of bound percentage is much higher than for the other streams. Can you explain this?

The interval is narrower because the model fit was better, and the residuals were lower. This led to a lower proportion of the observed values being outside the prediction interval.

L389: Consider using the same y axis limits for all plots. Like this, it is difficult to see the differences between the streams. At least the zero line should be visible in all plots.

We could not use the same y limits for all plots, as there are large differences between the discharge vales per stream. We will add the zero line.

L422: Which physiographical statistics did you analyse? Please provide details. I guess you did not analyse crop types (e.g. fraction of arable land (without grassland) on the total catchment area)? Could this also be a reason for the difference?

That is a good point, thanks for bringing it to our attention. We will provide details of the characteristics we analysed (e.g. stream and road density), and mention that crop types might contribute to explain these differences.

L443: Where do you show in your study that model resolution is important for your results? I don't think that you can conclude this from your study.

We are simply stating that the model spatial resolution needs to be sufficiently fine to represent connectivity features and processes. If we were using 30 m resolution data, we would not even be able to perform this study. For instance, we would not be able to represent the roads or the grass strips.

L444-445: In L285 you state "Hence, modelled hillslope yields and suspended loads are not fully commensurable, and we did not focus on a rejectionist framework for model testing." Here, you state that "soil redistribution rates and patterns are intrinsically linked to linear

features". The strength of the latter statement does not really fit to the caution you demand in the first statement. Therefore, you should reformulate this sentence.

We partially agree with this comment. As we explained above, our conclusions are not solely based on the comparison between model outputs and the catchment sediment loads. We will rephrase, referring to field-based studies that should allow us to state:

"In a wider context, our study has demonstrated how structural sediment connectivity patterns can be investigated with a conceptual model as WaTEM/SEDEM, provided that model resolution is sufficiently fine to represent relevant features and processes. In agricultural catchments of the Swiss Plateau and likely in other patchy landscapes, soil redistribution rates and patterns are intrinsically linked to linear features (Alder et al., 2015; Ledermann et al., 2010; Prasuhn, 2020; Remund et al., 2021). Hence, in order to provide relevant system descriptions, soil erosion models applied under similar conditions must be able to represent linear features and landscape patchiness".

L472-475: I very much like this part. In contrast to the discussion, I feel that here the strength of statements fits together with what you did in your work and with the related uncertainties.

Thanks, we appreciated your constructive criticism throughout the text, which helped us improve our paper.

L478: the effects -> the potential effects

We will rephrase.

L484: Would you not rather recommend a proper validation of you model before upscaling it?

This is precisely what we tried to recommend, sorry if it was not clear. We will rephrase it to make it more precise.

**Technical corrections**

L11: "In particular": Seems to be the wrong transitional phrase. Please rewrite.

"In particular" is a transitional phrase used to illustrate or explain an idea, which was our goal here.

L13: "grass-buffer-strips": Is that the correct term/correct spelling? (Revise in whole manuscript.)

Thanks for noticing. We checked and this should not be hyphenated, we will correct it throughout the manuscript.

L31: increase -> increases

Thanks, this will be corrected.

L44: infra-structure -> infrastructure

Thanks, corrected.

L75: weekly -> weakly

Corrected.

L89: Baldegg Lake -> Lake Baldegg (revise in whole manuscript; see for example https://doi.org/10.1039/D0EM00317D)

L97: of the canton of Lucerne. (revise in whole manuscript)

L111: field-blocks -> field blocks (Revise in whole manuscript.)

All hyphenations and place names will be revised.

L114: Consider just writing $km^{-1}$

We think $km\ km^{-2}$ is more intuitive.

L120: Elevation -> The elevation

Updated.

L141: water discharge values -> discharges

Updated.

L194: Make a reference from this. No URL directly in the text. Check if there's a permanent identifier/URL.

Not sure what you mean here. In any case, all formatting issues will be dealt with.

L221: Wrong table referenced.

Thanks, updated to Table 2.

L235-238: Difficult to read. Rephrase.

We would appreciate more guidance here. What did you find difficult to understand?

L292: This can be easily visualised -> This is shown

Updated.

L257: Is it a hydrological or hydraulic shortcut?

Corrected to hydraulic throughout the manuscript, thanks.

L306: increased in -> by or to

Thanks, this will be corrected.

L372: In Figure 7, you write "short-cut", while in the whole manuscript you wrote "shortcut". Please adapt.

We will correct this in the figure, thanks.

**References**

Alder, S., Prasuhn, V., Liniger, H., Herweg, K., Hurni, H., Candinas, A. and Gujer, H. U.: A high-resolution map of direct and indirect connectivity of erosion risk areas to surface waters in Switzerland-A risk assessment tool for planning and policy-making, Land use policy, 48, 236–249, doi:10.1016/j.landusepol.2015.06.001, 2015.

Borrelli, P., Alewell, C., Alvarez, P., Alexandre, J., Anache, A., Baartman, J., Ballabio, C., Bezak, N., Biddoccu, M., Cerdà, A., Chalise, D., Chen, S., Chen, W., Maria, A., Girolamo, D., Desta, G., Deumlich, D., Diodato, N., Efthimiou, N., Erpul, G., Fiener, P., Freppaz, M., Gentile, F., Gericke, A., Haregeweyn, N., Hu, B., Jeanneau, A., Kaffas, K., Kiani-harchegani, M., Lizaga, I., Li, C., Lombardo, L., López-vicente, M., Lucas-borja, M. E., Märker, M., Matthews, F., Miao, C., Modugno, S., Möller, M., Naipal, V., Nearing, M., Owusu, S., Panday, D., Patault, E., Valeriu, C., Poggio, L., Portes, R., Quijano, L., Reza, M., Renima, M., Francesco, G., Rodrigo-comino, J., Saia, S., Nazari, A., Schillaci, C., Syrris, V., Soo, H., Noses, D., Tarso, P., Teng, H., Thapa, R., Vantas, K., Vieira, D., Yang, J. E., Yin, S., Antonio, D., Zhao, G. and Panagos, P.: Science of the Total Environment Soil erosion modelling : A global review and statistical analysis, , 780, doi:10.1016/j.scitotenv.2021.146494, 2021.

Ledermann, T., Herweg, K., Liniger, H. P., Schneider, F., Hurni, H. and Prasuhn, V.: Applying erosion damage mapping to assess and quantify off-site effects of soil erosion in Switzerland, L. Degrad. Dev., 21, 353–366, 2010.

Panagos, P., Borrelli, P., Meusburger, K., Alewell, C., Lugato, E. and Montanarella, L.: Estimating the soil erosion cover-management factor at the European scale, Land use policy, 48, 38–50, doi:10.1016/j.landusepol.2015.05.021, 2015.

Remund, D., Liebisch, F., Liniger, H. P., Heinimann, A. and Prasuhn, V.: The origin of sediment and particulate phosphorus inputs into water bodies in the Swiss Midlands – A twenty-year field study of soil erosion, Catena, 203(March), 105290, doi:10.1016/j.catena.2021.105290, 2021.

Schmidt, S., Alewell, C. and Meusburger, K.: Mapping spatio-temporal dynamics of the cover and management factor (C-factor) for grasslands in Switzerland, Remote Sens. Environ., 211(April), 89–104, doi:10.1016/j.rse.2018.04.008, 2018.

---

## Author Response (AR1)

Dear Genevieve and referees,

Apologies for the delay in our response and many thanks for your patience. Below we answer to all your questions, comments, and suggestions. For clarity and improved visualisation, the reviewer comments are shown from here on in black. The authors' replies are in blue font below each of the reviewers' statements. The changes in the revised manuscript are displayed in green. Line numbers refer to the revised tracked manuscript.

Please note that while editing figure 5 following the referee comments, there were some changes in the mean squared error (MSE) values calculated by the random forest. This is because the randomness of the tree building. We have now used a set seed to ensure the reproducible random objects. Importantly, these changes in the MSE values were slight and did not at all affect the interpretation of the results.

**Reviewer #1**

**General Comments**

The paper is excellent in terms of organization, writing, and content. The results and figures are convincing, and I am particularly excited about how the authors utilize changes in model structure to address sediment connectivity, which is timely and quite important in terms of advancing watershed sediment simulations. Overall, I have very minor comments regarding some clarifications in a few instances and I believe that a few statements made by the authors should be relaxed a bit. Additionally, I would suggest including a brief paragraph at the end of the discussion regarding limitations of the WaTEM/SEDEM approach, and how we can further move to improve the spatial and temporal resolution of sediment connectivity simulations.

Thanks again for reviewing our paper, we highly appreciated the input. Below we respond to all your specific comments. Moreover, we have included a paragraph in the discussion further highlighting the limitations of the WaTEM/SEDEM approach and some thoughts on how to improve sediment connectivity simulations.

**Specific Comments:**

L 32: Perhaps you can also mention that the paper is, for the first time to my knowledge, advancing tools to assess connectivity by quantifying structural uncertainty within the sediment

simulations (not referring to *structural* connectivity here, by the way, just how there are inherent uncertainties within how the model is configured to predict fluxes/loads).

Thanks for pointing this out. We highlighted how the quantification of structural model uncertainty is an important and novel aspect of our work, throughout the text (L23, L36, L237-240, L319)

L 159-167: It would be helpful within the text to tell readers the temporal resolution of the model. It seems like it's yearly according to the RUSLE equations but could be clarified.

Apologies for this omission. Indeed, the model is operating in a yearly time-step, as we now mention in lines 188-188.

L 180: Is this of the individual pixel or along the slope length?

The slope gradient and the transport capacity are calculated per individual pixel (L195).

L 188: Does this include bank erosion?

We included an explicit mention to bank erosion here, as this process is also not simulated by the model (L210).

L 188-190: If this is a yearly model then perhaps this statement can be slightly relaxed... For example if the system is known to not aggrade or degrade over longer-term (decadal) timescales then instream erosion and deposition are approximately in equilibrium and so I would not be as concerned with the instream.

We completely agree, thanks for pointing this out. We rephrased to:

"Since WaTEM/SEDEM does not represent gully and bank erosion or in-stream erosion and deposition processes, any comparison between modelled sediment yields and catchment-outlet sediment loads must be interpreted with caution. However, in catchments where rill and interrill are the main overland erosion processes, and assuming a state of long term fluvial quasi-equilibrium, the outlet sediment loads should be at least comparable to the model outputs, even if not fully commensurable". (L212-216)

L 205: How did you decide 1200? Is this enough? Sometimes people will utilize 100,000 monte carlo simulations. I'm not saying that you need to run the model for more realizations, just a bit more justification please.

In all honesty we do not have a strict justification for the number of iterations in the Monte Carlo simulation. We understand 1200 model realisations was enough to explore the parameter space for the purpose of the sensitivity analysis. If we wanted to sample the parameter space exhaustively in a rejectionist approach, then we would likely need a higher number of simulations considering the number of parameters.

L 205-208: Right, the typical approach is to calibrate the model and along the way assess sensitivity/uncertainty such that sensitivity/uncertainty of the model is addressed within solution spaces that are plausibly behavioral. I'm not rejecting your approach by any means, but perhaps some additional acknowledgement of the traditional approach and how you are slightly deviating here could be helpful to readers. Some readers might question why you present realizations that will not adequately describe the sediment load/flux in the system. L 208: *model assumptions* – I would clarify you are making assumptions about the structure of the model, so quantifying structural uncertainty.

Precisely – with the sensitivity analysis we did not aim to identify the behavioural parameter space, but rather to understand how the model responds to the different structural assumptions. We will made the following changes in this paragraph:

"Our model application consists of a global all-at-a-time sensitivity analysis, as described by Pianosi et al. (2016). That is, we performed a Monte Carlo simulation to explore the variability of the whole parameter space, and all input factors were sampled simultaneously for each model realisation (n = 1200). The framework is similar to an uncertainty analysis, except in this case we did not focus on locating the parameter space which produced behavioural model realisations. Instead, we concentrated on apportioning sources of uncertainty to different model input factors, aiming to rank their contribution to the variability of the response surface (see Pianosi et al., 2016 for a review on sensitivity analysis). This should allow us to identify parameters and model assumptions that have a greater impact on the manner with which WaTEM/SEDEM describes sediment connectivity in the Baldegg catchment. In particular, the analysis of different assumptions about the structure of the model should provide a connectivity assessment based on the apportionment of the structural uncertainty withing the simulations.

To the best of our knowledge, this is the first time the analysis model structural error is incorporated into sediment connectivity research." (L228-240)

L 238: can you clarify why a value for *Pcon* wouldn't be applied everywhere in the catchment, but instead for just the forest and buffer strips? What if there is disconnectivity from microtopography in the roadside ditches, for example? Again – I'm not asking for additional analyses, just a sentence or two for clarification and that you *might* parameterize this other places in the watershed if you had overt reason to.

This is a great point, thanks for bringing it to our attention. The parcel connectivity parameter was originally developed to represent the extent with which water and sediment transport is reduced at parcel borders in case the downslope patch is composed of forests or grasslands. We clarified this in the text (L273-274). We completely agree that the parameter could be incorporated in other places, and we will mention this in the model limitations/improvements section you suggested.

L 255-259: can you please add a sentence that details the difference between scenario two and three? The way I understand it is that in scenario three sediment deposition does not occur on the road or in swales/ditches along side the road, but deposition can still occur downstream, for example in between the road and the stream network. In scenario two sediments are automatically connected to the stream, correct?

That is absolutely correct. We now state:

"For this scenario, deposition will never occur on road cells, however sediments can still be deposited on downstream patches, before reaching the stream network". (L299-300)

L 270: Again I might suggest using the word *structural* uncertainty of the model.

Thanks, we will included a mention to structural uncertainty here (L310).

L 275-276: This was a bit confusing to me. L 306: What is the mean squared error in relation to? The yearly predicted sediment load and the yearly average sediment load from the rating curves? Please clarify.

Apologies for this confusion. The random forest analysis was used to predict the WaTEM/SEDEM simulations of hillslope sediment yield, based on the parameter values that

were sampled for each iteration. The mean squared error is calculated from the RFA predictions and the WaTEM/SEDEM simulations. The increase in error due to the absence of a variable is used to rank its importance. We will clarify these points in the methods and the results (L315-317, L349-351).

L 323: Fig. 6g – what about interrill erosion?

Thanks for noticing this, we included interrill erosion in the figure legend.

L 323: Fig 6b,c; L 333: Is it worth showing land use for all the details here?¨

This was indeed missing, thanks for noticing. We included some text in the figure showing where the arable land and the forest are located (Fig 6b).

L 374: Perhaps you can say in the caption that the short-cut generally overlaps the IQR better than the other 2 scenarios... this could help readers quickly interpret the figure.

Thanks, we included this in the caption of figure 7.

L 385: Out of bound percentage – is this a fraction or a percentage what is presented in the table?

Apologies for this mistake, indeed we were presenting the fraction. This has been corrected in Table 4.

L 388: It would be great if we could see at this same time scale how SEDEM was performing… but I think this is just a limitation of the model since it runs at a yearly scale, correct?

Yes, exactly. For this reason, we only made comparisons with the average yearly loads.

L 424: Perhaps also the rating curve is underestimating the load, as you previously mentioned? Which would improve the performance of simulations with respect to the short cutting, correct?

We are not sure this would explain this pattern, as the curves are probably underestimating the actual loads for all streams – not just for the Höhibach. In any case, we explained now how underestimation would probably improve the performance of the shortcut simulations (L436).

L 451: I believe Mahoney et al., 2018 talks about importance of road networks a bit in the USA.

Thanks, we included the reference (L511).

L 463-465: I think this last sentence should be relaxed a bit... quantifying all of the sources of uncertainty due to observation data, model input data, model output data, parameter uncertainty, etc. etc. is quite the undertaking. In fact, in my opinion, it might be an impossible task. Does this invalidate the use of models, however? In my opinion, no, it does not. We can still discern important information from models even though we don't account for 100% of uncertainties. It ultimately will depend on what questions we are trying to answer with the model and what the model is attempting to do, which can be equally as important as quantifying certain uncertainties in my opinion.

We completely agree that accounting for all uncertainty is impossible. What we were trying to convey here is that soil erosion and numerical connectivity models are highly uncertain. This uncertainty stems from multiple unknowns about the modelled phenomenon, the input data, and the forcing data. Our opinion is that neglecting such uncertainty makes it difficult to provide meaningful insight based on the modelling. In any case, we see how our statement might have been too strict here, and we rephrased to:

"As recent studies have again demonstrated, investigating the uncertainty in model structures, parameter estimation, and observational testing data is crucial for advancing soil erosion modelling research (Benaud et al., 2021; Eekhout et al., 2021; Schürz et al., 2020)" (L526-530)

L 465-466: It would be nice if a paragraph on limitations of the modeling approach and future opportunities could be included. For example, while RUSLE is relatively easily implemented and approachable, it would be nice if the RUSLE approach was a bit more physically based. Additionally the RUSLE approach limits the temporal resolution of the model, so seeing event- and seasonal-scale connectivity seems a bit limited. Furthermore, the advanced geospatial data that facilitates this novel connectivity modeling is wonderful, and can help to elucidate hotspots of connectivity. Additionally there is recent sentiment to move towards high-temporal resolution models to quantify hot-moments of connectivity. The yearly timescale inherent to the RUSLE approach perhaps is underserving this sentiment.

We completely agree. We think a model like WaTEM can be useful to explore structural connectivity patterns, but much more dynamic models are needed to quantify these hot-moments of connectivity, and to get a better grasp of the functional connectivity of the system. This is now discussed in lines 531-540.

L 466: I'd suggest perhaps emphasizing that exploring structural uncertainties in the model framework - and not just parameter uncertainties, as is the traditional method - allowed for advanced understanding of connectivity processes. This type of approach in my opinion is quite underserved in modeling work and should be considered in the future where high-resolution geospatial data is available.

Thanks for pointing this out. We emphasized the relevance of quantifying structural uncertainties throughout the manuscript

Technical Comments:

L 312: I'm not sure if the different colors are helpful here, maybe consider symbols?

The colours are just there to keep the theme of the graphs – the facet titles identify the scenarios. If it's ok, we would like to keep them. We did include the symbols now, however (Figure 5).

L 324: typo

Thanks, corrected.

L 364: typo, confusing

We rephrased to: "The comparison between WaTEM/SEDEM simulations and the tributary sediment loads revealed a larger overlap between the latter and the results from the 'road as shortcuts' scenario (Figure 7)" (L410-412).

**Reviewer #2**

**General comments**

This paper provides a modelling approach to analyse the effect of linear features on sediment connectivity in a mesoscale catchment. The authors put a lot of work into adapting an existing model such that it is able to account for these effects. I appreciate this effort and I think that this work is important for improving the understanding of sediment connectivity in agricultural catchments. However, I have some major points of criticism which are the following ones:

We highly appreciate the time and effort put into reviewing our manuscript. We specifically appreciate the attention to detail and the discussion about what conclusions can be drawn from the study. We address all your comments below.

The authors state themselves that a comparison between their model results and the measuremed sediment loads in the river should be performed with upmost caution. However, in the results and in the discussion they still make strong conclusions based on exactly such a comparison. The authors should therefore reformulate the discussion such that it reflects this uncertainty better.

We apologise if this were conveyed: we tried to be very cautious about the conclusions drawn simply from the comparisons with the sediment loads. Overall, the importance of linear features for regulating sediment connectivity has been extensively documented by field observations and modelling approaches in Central Switzerland (Alder et al., 2015; Ledermann et al., 2010; Remund et al., 2021). Results from our study corroborate these observations, both due to the sensitivity analysis and the comparison with the sediment loads, which we understand were cautious (or even overly cautions considering some of the comments from reviewer #1). As we state in the manuscript, the comparison between outlet sediment loads and model outputs provide an estimate of the plausibility of the simulations.

In any case, we reformulated the sentences you mentioned in the specific comments in order to be more careful about any potentially overreaching conclusions. This is explained below in our reply to your specific comments.

The comparison between the model results for the different scenarios and the measured sediment loads does not make sense for me in the way it is done currently. In their current analysis, the authors ignored important factors influencing the sediment connectivity in the catchment: Firstly, as I understand it from the manuscript, they treated all grassland areas like arable land areas in the model. Since around a third of the catchment area is covered by grassland, this leads to a large overestimation of sediment loads.

Here we believe there has been a misunderstanding, which is entirely our fault. Indeed, we did not separate grasslands from croplands (reasons why are explained in the specific replies). However, we considered this uncertainty in the parameterisation of the CP factor, which had the lower limits of their "prior" distribution stretched to include typical values for permanent grasslands in Switzerland. This is now explicitly stated in lines 245-247. Similarly, the Ktc high

parameter was sampled from 0 to 200 m to cover a wide parameter space. Hence, some model realisations will indeed overestimate the erosion rates, whilst others might underestimate them. Since we did not focus on calculating exact erosion rates for the catchment, but rather on assessing the influence of linear features and landscape patchiness in sediment connectivity, we do not see any major limitations with our approach. Particularly since all this uncertainty is incorporated into the model outputs.

Secondly, the authors assumed a two-meter grass buffer strip around all agricultural plots. The authors state that they don't know the real width of buffer strips in the field and that they therefore use a value of two meters for testing the sensitivity of the model. Even though the sensitivity analysis showed that the buffer width had a large influence on the model results, the comparison between modelled and measured sediment loads is only done for the two-meter scenario. Also the conclusions are only drawn based on this scenario. Even though the authors state that the two-meter scenario is more realistic than the scenario without a buffer, this value contradicts the values reported by other studies (e.g. Alder, 2015; Remund, 2021) and the legal requirements. Although this point is addressed in the discussion, I am missing a proper justification why the measurements are only compared to the two-meter scenario.

We assumed a 2 m width to test the sensitivity of the model to the presence of buffer strips. The value was pragmatically chosen based on the spatial resolution of the model input data – apologies for not mentioning this in the methods. In addition, we understood it was better having one fixed value for testing such sensitivity, than having multiple values which would anyway be uncertain. To clarify, our sensitivity analysis did not demonstrate the model was sensitive do the buffer *width*, as we only tested one value. We think testing additional values goes beyond of the scope of our research.

Now, if the buffer strips alongside roads in the catchment (which are at least 0.5 m wide according to legislation, but highly variable in width) are completely ineffective, then the scenario without grass strips would possibly be the most appropriate representation of the system – at least considering the capacity of model outputs to mimic the data. However, the strips would still be there in the fields, and this was our rationale when we stated that the scenario with strips "more closely represent[s] the actual structure of the agricultural fields in the Baldegg catchment". As all results are summarised in Table 3 of the manuscript, and the scenario without strips shows the same pattern for the different road connectivity assumptions as the scenario with the strips – leading to the same conclusions – we thought it would be

interesting, for concision and clarity, to focus the discussion on the latter. In any case, we have now included the data for the scenarios without grass strips into Figure 7. As you can see, this does not change the interpretation of our results.

Thirdly, in their "shortcut" scenario, the authors assume that all roads and farm tracks are drained with shortcuts. (At least, this is how I understand it from the manuscript.) I expect that a major part of the roads in the Baldegg catchment is actually not drained by shortcuts. Therefore, it makes sense to use the current "shortcut" scenario in a sensitivity analysis, but not as a realistic scenario. Compared to reality, the current "shortcut" scenario is expected overestimate the real sediment loads. Consequently, even though the "shortcut" scenario is most similar to the measurements, this is possibly simply caused by a bias in the model input. Even though I agree that roads and shortcuts are in fact important for sediment transport, I think this cannot be concluded from the current analysis. To state that "roads behave as conduits for sediment transport in the catchment", as it is done in L474f, it is inevitable that the scenarios are revised such that they reflect the reality in the catchment better. (At least for the first point mentioned. Second and third point may also be discussed.)

We completely agree that in reality only a certain portion of the roads will drain the sediments directly to the stream network (note that we never used the word "realistic" to describe any of the scenarios). However, as we explained above, we used quite a wide range of parameter values and different scenarios to examine *how things could happen* in the catchment. Still, the model was only able to provide reasonably comparable results to the outlet sediment loads if we assumed that roads behave as conduits for sediment transport. And this is considering the large uncertainties in both the sediment rating curves and the model outputs. These findings, in combination with the results from the sensitivity analysis, and the multiple studies that report similar patterns for other sites in Switzerland (as reviewed in the discussion), should allow us to state the following in the conclusion (L529-538):

"Our results demonstrated that assumptions about road connectivity were by far the most important factor for modelling sediment transfer in the Baldegg catchment. Moreover, the comparison between extensive model simulations and sediment rating curve calculations indicated that roads and hydraulic shortcuts are likely to behave as conduits for sediment transport in the catchment. Hence, representing road connectivity is crucial for modelling sediment transfer from hillslope to water courses in this agricultural catchment of the Swiss Plateau, and potentially in other areas with a dense road drainage system. Moreover, our results

further highlighted the effects of linear structures and landscape patchiness on sediment connectivity."

In addition, if the modelled sediment loads were highly overestimated, as you have hypothesised above due to a potential bias in the CP parameterisation of grasslands/croplands, then it would not make sense that the shortcut scenario showed a better fit with the sediment rating curve calculations. In this case, the shortcut scenario would exhibit much higher sediment loads than the measurement-based estimates, as both erosion rates and sediment connectivity would have been overestimated.

The authors state that the catchment is representative for the Swiss plateau. However, they do not further elaborate on this. Other studies, however, rather suggest that the catchment has a low shortcut connectivity compared to other catchments in the Swiss plateau (see comment to L98). The authors should improve on putting their analysis in the right context.

Thanks for pointing this out. We removed any mentions in the manuscript about the catchment being representative for the Swiss Plateau. Indeed, that cannot be affirmed, and it is not our objective to provide a representative case study for this part of Switzerland.

Several specific comments (see below) should be addressed to improve clarity and reproducibility of the study. Additionally, the manuscript should also receive some revisions regarding language and correct spelling.

From our understanding, the comments on reproducibility are mostly related to software and package versions, and to methodological details regarding data preparation. These were addressed accordingly, as we explain in the replies to your specific comments below. In addition, we have corrected the typos and spelling errors you highlighted. We would be glad to send the manuscript over to a native speaker for review if the editor deems necessary.

**Specific comments**

L36: Talking about a "continuous displacement of small amounts" is wrong here. The discplacement varies strongly between events and years, as you also state below.

We rephrased to: "Rainfall events on sloped surfaces continuously displace soil particles, which are transported downslope as sediments" (L40).

L47: Rephrase.

We would appreciate more guidance here about what needs rephrasing and why.

L56: You should also add the most recent publications here, e.g.: https://doi.org/10.1016/j.catena.2021.105290

Thanks, that was a very appropriate reference and it has been included to the text (L61).

L66: "assuming they are able to explicitly take connectivity into account": Difficult to understand. Please write this more clearly.

Apologies if that was not clear. We rephrased to: "These usually rely on high-resolution process-based models, assuming they are able to represent connectivity dynamics" (L72).

L73: with a size of few square kilometres

Thanks, we updated the text accordingly (L79).

L79: You state above that one major issue of erosion models is the uncertainty of input data. Then you state that you used a high resolution dataset (2x2m DEM). However, for whole Switzerland, a 0.5x0.5m DEM is freely available in the same quality as the 2x2m DEM. You still used the 2x2m DEM. Why did you not use the higher resolution model?

In such high resolutions (0.5 m), the influence of the microtopography becomes much more prominent, and we understood this conceptual model would not be the most appropriate for handling such features. We would like to highlight however that the 2 m resolution we are working with is much higher than what is usually employed in erosion modelling research, in particular at catchment/mesoscale (see Borrelli et al., 2021).

L98: Here you state that the Baldegg catchment is *patch*y and *representative* for the Swiss Plateau. Below, you only elaborate on the patchiness, but not at all on the representativeness. Either elaborate on the representativeness below or use another word here. Schönenberger et al. investigated two catchments in proximity or even inside your catchment. Compared to the distribution in the Swiss plateau, these catchments however rather seem to have a low shortcut connectivity. This indicates in my opinion that also your catchment is rather on the lower side with respect to shortcut connectivity.

We agree that we did not elaborate on the representativeness of the catchment and ultimately this is not our goal here. Hence, we removed any reference to the catchment being representative for the Swiss Plateau.

L103: In Figure 1c, you use the term "infrastructure", here you use the term settlements. It is unclear how these two terms differ and what the term infrastructure means. Please use consistent terms here. Additionally, how did you treat the areas of roads? Did you include them into the settlement area? Or are they included in the agricultural land/forest area?

Apologies for this inconsistency. Roads and settlements areas were calculated as part of the infrastructure, as we now state in line 114.

L117: Are tile drainage only located in water accumulation zones? (What are water accumulation zones? Are the determined based on topographic index, slope?)

By definition tile drainage is found where there is excess water. Upon reflection we found this information superfluous, and we will remove it from the revised manuscript.

L120: How did you determine these slopes? Which elevation model? The maximal slopes are strongly depending on the model used. You are referring to Figure 1b. However, the slope is not visible in Figure 1b.

Figure 1b is referred for altitude "higher altitudes are found in the eastern and western sides of the catchment (Figure 1b)" in line 134. Slope was calculated with the same DEM used for the model.

L122: "in this case formed by the retreat of the Reuss Glacier in the south to north direction (~18,000 years BP)" -> Not important. Consider removing.

We think this information contributes to the description of the study site and would appreciate if we could keep it.

L127: MeteoSwiss -> Please add reference.

Apologies for this lapse, the reference has been included.

L132: What is approximately? Provide the range of numbers of grab samples taken.

Apologies, we have corrected to "on average 275 grab samples were taken from each tributary" (L147)

L133: What is "opportunistical sampling"?

This has been removed and further information on the sampling was provided (L144-149).

L136: Why 2020? Above you write that you only sampled till 2019.

Apologies, we meant until the end of 2019 (L152).

L144: What is k? I guess the covariate ID. This should be written explicitly. Additionally, in contrast to Vigiak & Bende-Michl, you are only using the first five covariates, but not the long-term trend covariates (6 and 7). Why?

Thanks for noticing this. Now we explicitly mention that k is the covariate identity (L164). We did not use the long-term trend covariates because they were either not significant or did not improve the models, which makes sense considering the timescale of our analysis.

L149: First column of table: Remove the word "is". Also the word "water" is not really needed. Second column of the table: This is not the quadratic term of $Q_i$, but of $x_{2,i}$.

Many thanks again for noticing these errors, they have been corrected in Table 1.

L156: You are addressing the variance in sediment concentrations extensively. However, you are not addressing the uncertainty in daily discharge at all. Why?

Essentially because the uncertainty in the sediment load calculations is much larger than for water discharge. We have on average 275 measured sediment concentrations per tributary, which need to be extrapolated for 10 years. On the other hand, daily discharge measurements are available.

L165: Shortly explain why you only focused on water erosion.

We chose to focus on the water erosion instead of tillage because the latter is not relevant for the type of connectivity processes we are investigating (L184-187)

L193: "usually implemented" -> Rephrase. (It is either implemented or not. Possibly, you could write something like that this version is often used.) Provide references where this version is used.

Thanks, this was rephrased to "WaTEM/SEDEM is implemented as a user-friendly GUI […]" (L218).

L199: R version? L200: SAGA version?

Versions are now supplied in the reference list and code.

L201: The code does not contain any information on the versions of the packages used. To make sure that the code can still be used in the future, you should at least provide information on the package versions used. Consider also using tools like packrat, checkpoint or docker. To make the code useful, you should also provide examples of input files.

Thanks for the package recommendations. We included information on package versions in the code – apologies for this missing information. As we stated in the data availability section, we will upload the model input data to Envidat if the manuscript is accepted.

L219: It remains unclear how you derived the land cover map. The reference is not shown in the reference list. Therefore, I don't understand if you used a vector dataset that you rasterized yourself or if you used a raster dataset provided by Swisstopo. What does the resolution "1:25:000" mean in the latter case? If you used vector datasets: How did you deal with point and line features, e.g. roads or hedges? Did you assume widths for roads and hedges?

We are terribly sorry for missing Swisstopo in the reference list. Indeed, we rasterised the land cover vector data (Swiss Map Vector 25 BETA, scale 1:25,000) to a 2 m resolution. The roads were firstly converted from lines to polygons with a buffer, considering their width (which is informed in the vector map). Hedges and tree lines were already represented as polygons in the vector map. We made it clearer in the manuscript that we rasterised the vector map ourselves (L2250).

L220: The statement that spatially distributed crop statistics are unavailable is wrong. There is a plot-resolution crop dataset from the canton of Lucerne available freely for the whole canton (and accordingly for the whole catchment) for the year 2019. Why was this dataset not used? (CP and $K_{TC}$ depend on the crop and you reported them to be the most sensitive model

parameters.) Lavrieux et al. state that one third of the agricultural area consists of permanent grassland. Therefore, I expect this decision to have a large influence on your results, leading to an overestimation of erosion. You should address this point at least statistically for each of the five subcatchments analysed and for the full catchment. (e.g. look at the fractions of grassland and reduce the estimated amount of sediment load accordingly). Alternatively, you could also do a spatially explicit analysis.

In all honesty we were not aware of the availability of such data for the canton of Lucerne. We inquired the canton about it, and the geodata would indeed be available. However, this would cost 4260 CHF.

In any case, as we stated before, we apologise for this misunderstanding regarding the grasslands. As we explain in the revised manuscript (L255-257), "The minimum CP values were particularly reduced to include typical values for permanent grasslands in Switzerland (~0.01) (Schmidt et al., 2018b)". This lower limit of the CP parameter is also analogous to the lowest value recommended for permanent grasslands in Europe (0.01-0.08) (Panagos et al., 2015). Similarly, the Ktc high parameter was sampled from 0 to 200 m to sample a wide parameter space. Hence, some model realisations might overestimate the erosion rates in the catchment, whilst other realisations will likely underestimate them. In either case, what we can see with the model results is that whatever values we sample for the CP or Ktc parameters, the assumptions about connectivity have the highest impact on the model results. In summary, we have already addressed the land use issue 'statistically', as you put it.

L226: Why did you use the 2x2m resolution DEM? (see comment to L79) How did you process the DEM, e.g. sink filling?

The choice of DEM resolution is explained in the comment to L79. The DEM was sink filled by the Wang & Liu (2006) method, which is implemented in SAGA.

L227: In the $K_{TC}$ column, consider indicating the land use classes belonging to "high" and "low" (e.g. in brackets). This would make it much easier to read.

Thanks, we included the information in brackets (Table 2).

L230: Specify that this relates to the maximum CP factor.

This is clear now as we state the minimum value was selected based on the permanent grasslands (L255-257).

L233f: You talk about forests and grass buffer strips in the land cover map. However, the reader is missing were you explain the derivation of forests and grass buffer strips. Consider stating that you are explaining this below.

We now state we will explain the grass buffer strips below (L267).

L246: How much wider? Refer to the article in the corresponding legislation directly, instead of Alder et al..

Sorry for the missing information. We rephrased to: "The extent of the buffer-strips in reality is quite variable, and generally wider at forest and river vicinities (3 – 6 m), as required by law in Switzerland (Alder et al., 2015)." (L281-282)

We would like to keep the reference to Alder et al., which provides this and other information about grass buffer strips in Switzerland in English.

L249: What were your assumptions on buffer strips along hedges? Did you also use a 2m buffer? Or a buffer corresponding to the legal requirements? How did you treat tree lines?

We used a 2 m buffer along the hedges and treelines, which were rasterised from the land cover map mentioned previously.

L250: As mentioned in the comment to L219, I don't really understand how you derived the road areas.

The road widths are provided in the land cover vector data (Swiss Map Vector 25 BETA). We used these widths to perform a buffer around the road lines. Next the polygons were converted into a 2m resolution raster (same resolution as the DEM).

L252: What is "infrastructure"? Was this also derived from the land cover map?

Infrastructure includes roads and settlements. This is now informed in the text (L114)

L253: If roads act as sinks, why is this related to field drainages? From the text, I don't understand where you assume the sediments to be trapped. On the road? In the drainage system?

In sludge collectors? The scenario seems to make sense for me, but you should specify your assumptions more clearly.

We rephrased to "This represents a scenario in which roadside ditches and the road drainage system trap most sediments and partly diverge runoff to wastewater treatment plants" (L290-291).

L256: Did you assume here that all roads are acting as shortcut? Or only a part of the roads?

We assumed all roads behave as shortcuts. We now mention how this could be improved if the location of the shortcuts and the extent with which they are connected to surface waters were known, as in Schönenberger and Stamm (2021) (537-540).

L262f: How many directions?

Information on the multiple flow direction algorithm implemented in SAGA can be found in Freeman (1991) and Quinn et al. (1991). It allows for divergent flow paths, differently to the typical D8 approach.

Freeman, G.T. (1991): Calculating catchment area with divergent flow based on a regular grid. Computers and Geosciences, 17:413-22.

Quinn, P.F., Beven, K.J., Chevallier, P. & Planchon, O. (1991): The prediction of hillslope flow paths for distributed hydrological modelling using digital terrain models. Hydrological Processes, 5:59-79.

L266-268: In my opinion, some (or probably all) of the additional packages (e.g. "doParallel", "foreach") are not worth mentioning here, as they are only used to speed up the calculation process, but not important for reproducibility of your work.

Thanks, we removed these references.

L278: What does RFA stand for?

Apologies, RFA stands for random forest analysis (L314).

L280: Version?

Information on package versions is provided in the reference list.

L299: Do I understand correctly that Mg yr$^{-1}$ means tons per year? Probably write tons instead.

The megagram (Mg) is equal to a metric ton. It is common approach to express erosion and sediment transport rates with such unit.

L299: For me, the differences between GS and NGS scenarios are not well visible in the plots. Consider making this better visible, e.g. by adding a moving average per category or something similar.

We think a moving average would not the best choice here, as the idea with the scatter plots is also to display the spread of the response surface.

L317-L318: How did you quantify this for the whole catchment? Just by eye? If yes, you should provide the respective plots (e.g. in the appendix). Otherwise, can you provide a quantitative assessment?

Figure 6b, which we refer to, provides an example of the mentioned depositional patterns. In addition, increased within-field deposition rates are expressed quantitatively in Table 3.

L324: In Figures 6b, 6c, 6e, 6f, and 6g, arrows indicating the flow direction would help to understand the plot better.

Thanks for the suggestion, we included arrows indicating the flow direction (Figure 6).

L358: In my opinion, Table 3 would be understandable easier if you would write 25%, 50%, and 75% instead of Q1, Q2, Q3.

Thanks, but in this case, we would like to keep the quartile numbers in the table header.

L366f: As I understand from L244ff, you say that you don't really know what the real widths of buffer strips are in the catchment. It therefore makes sense to me that you use a fixed width of 2m and use it for testing the model sensitivity by running two scenarios – one with and one without the buffer strip. However, in L366 you now state that the 2m scenario is more realistic than the "no buffer" scenario. I agree that a 2m scenario is probably the more realistic scenario along forests. However, I expect the effect of a grass buffer along forests to be small as sediments are trapped by forests anyways. Along roads, I doubt that the 2m scenario is more

realistic than the "no buffer" scenario, since the legal requirement is only 0.5m (as you write in L430). (However, I might be wrong with these doubts.) Since I expect the buffer width along roads to be much more important for your model results than the ones along forests, I think you should report both scenarios here, or give a clear explanation on why you think or why you can show that the 2m scenario is more realistic.

Thanks for these considerations. As we stated previously, if the buffer strips alongside roads in the catchment, which are at least 0.5 m wide according to legislation, but highly variable in width, are completely ineffective, then the scenario without grass strips would possibly be the most appropriate representation of the system – at least considering the capacity of the model to mimic the outlet data. However, the strips would still be there, and this was our rationale to state that the scenario with strips "more closely represent[s] the actual structure of the agricultural fields in the Baldegg catchment". In any case, we have now included the data for the scenarios without grass strips in Figure 7.

L379-382: In the "Ron" stream, the 95% prediction interval seems much narrower than in the other rivers (Figure 8). Therefore, the observed values are mostly outside of this interval and the out of bound percentage is much higher than for the other streams. Can you explain this?

The interval is narrower because the model fit was better, and the residuals were lower. This led to a lower proportion of the observed values being outside the prediction interval.

L389: Consider using the same y axis limits for all plots. Like this, it is difficult to see the differences between the streams. At least the zero line should be visible in all plots.

We standardised the y axis limits (Figure 8). The zero line is not visible because of the log scale.

L422: Which physiographical statistics did you analyse? Please provide details. I guess you did not analyse crop types (e.g. fraction of arable land (without grassland) on the total catchment area)? Could this also be a reason for the difference?

That is a good point, thanks for bringing it to our attention. We now mention the characteristics we analysed (e.g. stream and road density), and explain that crop types might contribute to explain these differences (L477-478).

L443: Where do you show in your study that model resolution is important for your results? I don't think that you can conclude this from your study.

We are simply stating that the model spatial resolution needs to be sufficiently fine to represent connectivity features and processes. If we were using 30 m resolution data, we would not even be able to perform this study. For instance, we would not be able to represent the roads or the grass strips.

L444-445: In L285 you state "Hence, modelled hillslope yields and suspended loads are not fully commensurable, and we did not focus on a rejectionist framework for model testing." Here, you state that "soil redistribution rates and patterns are intrinsically linked to linear features". The strength of the latter statement does not really fit to the caution you demand in the first statement. Therefore, you should reformulate this sentence.

We partially agree with this comment. As we explained above, our conclusions are not solely based on the comparison between model outputs and the catchment sediment loads. We rephrased, referring to field-based studies that should allow us to state (L500-507):

"In a wider context, our study has demonstrated how structural sediment connectivity patterns can be investigated with a conceptual model as WaTEM/SEDEM, provided that model resolution is sufficiently fine to represent relevant features and processes. In agricultural catchments of the Swiss Plateau and likely in other patchy landscapes, soil redistribution rates and patterns are intrinsically linked to linear features (Alder et al., 2015; Ledermann et al., 2010; Prasuhn, 2020; Remund et al., 2021). Hence, in order to provide relevant system descriptions, soil erosion models applied under similar conditions must be able to represent linear features and landscape patchiness".

L472-475: I very much like this part. In contrast to the discussion, I feel that here the strength of statements fits together with what you did in your work and with the related uncertainties.

Thanks, we appreciated your constructive criticism throughout the text, which helped us improve our paper.

L478: the effects -> the potential effects

Upon consideration, we do not think the effects of linear features on sediment connectivity are potential – they have been observed in multiple field studies, as the ones we cite in the discussion. These effects have been highlighted by our work, due to the sensitivity of the model to the road connectivity assumptions and the presence of grass buffer strips.

L484: Would you not rather recommend a proper validation of you model before upscaling it?

This is precisely what we tried to recommend, sorry if it was not clear. We updated the text accordingly (L559-560).

**Technical corrections**

L11: "In particular": Seems to be the wrong transitional phrase. Please rewrite.

"In particular" is a transitional phrase used to illustrate or explain an idea, which was our goal here.

L13: "grass-buffer-strips": Is that the correct term/correct spelling? (Revise in whole manuscript.)

Thanks for noticing. We checked and this should not be hyphenated, we corrected it throughout the manuscript.

L31: increase -> increases

Thanks, changed to increases.

L44: infra-structure -> infrastructure

Thanks, corrected through the text.

L75: weekly -> weakly

Thanks, corrected.

L89: Baldegg Lake -> Lake Baldegg (revise in whole manuscript; see for example https://doi.org/10.1039/D0EM00317D)

L97: of the canton of Lucerne. (revise in whole manuscript)

L111: field-blocks -> field blocks (Revise in whole manuscript.)

All hyphenations and place names were revised.

L114: Consider just writing km$^{-1}$

We think km km$^{-2}$ is more intuitive and we would like to keep this.

L120: Elevation -> The elevation

Updated as suggested.

L141: water discharge values -> discharges

Updated as suggested.

L194: Make a reference from this. No URL directly in the text. Check if there's a permanent identifier/URL.

We removed the URL.

L221: Wrong table referenced.

Thanks, updated to Table 2.

L235-238: Difficult to read. Rephrase.

We would appreciate more guidance here. What did you find difficult to understand?

L292: This can be easily visualised -> This is shown

Updated to "this is illustrated".

L257: Is it a hydrological or hydraulic shortcut?

Corrected to hydraulic throughout the manuscript, thanks.

L306: increased in -> by or to

Thanks, changed to "by".

L372: In Figure 7, you write "short-cut", while in the whole manuscript you wrote "shortcut". Please adapt.

Thanks, this was corrected in Figure 7.

**References**

Alder, S., Prasuhn, V., Liniger, H., Herweg, K., Hurni, H., Candinas, A. and Gujer, H. U.: A high-resolution map of direct and indirect connectivity of erosion risk areas to surface waters in Switzerland-A risk assessment tool for planning and policy-making, Land use policy, 48, 236–249, doi:10.1016/j.landusepol.2015.06.001, 2015.

Borrelli, P., Alewell, C., Alvarez, P., Alexandre, J., Anache, A., Baartman, J., Ballabio, C., Bezak, N., Biddoccu, M., Cerdà, A., Chalise, D., Chen, S., Chen, W., Maria, A., Girolamo, D., Desta, G., Deumlich, D., Diodato, N., Efthimiou, N., Erpul, G., Fiener, P., Freppaz, M., Gentile, F., Gericke, A., Haregeweyn, N., Hu, B., Jeanneau, A., Kaffas, K., Kiani-harchegani, M., Lizaga, I., Li, C., Lombardo, L., López-vicente, M., Lucas-borja, M. E., Märker, M., Matthews, F., Miao, C., Modugno, S., Möller, M., Naipal, V., Nearing, M., Owusu, S., Panday, D., Patault, E., Valeriu, C., Poggio, L., Portes, R., Quijano, L., Reza, M., Renima, M., Francesco, G., Rodrigo-comino, J., Saia, S., Nazari, A., Schillaci, C., Syrris, V., Soo, H., Noses, D., Tarso, P., Teng, H., Thapa, R., Vantas, K., Vieira, D., Yang, J. E., Yin, S., Antonio, D., Zhao, G. and Panagos, P.: Science of the Total Environment Soil erosion modelling : A global review and statistical analysis, , 780, doi:10.1016/j.scitotenv.2021.146494, 2021.

Ledermann, T., Herweg, K., Liniger, H. P., Schneider, F., Hurni, H. and Prasuhn, V.: Applying erosion damage mapping to assess and quantify off-site effects of soil erosion in Switzerland, L. Degrad. Dev., 21, 353–366, 2010.

Panagos, P., Borrelli, P., Meusburger, K., Alewell, C., Lugato, E. and Montanarella, L.: Estimating the soil erosion cover-management factor at the European scale, Land use policy, 48, 38–50, doi:10.1016/j.landusepol.2015.05.021, 2015.

Remund, D., Liebisch, F., Liniger, H. P., Heinimann, A. and Prasuhn, V.: The origin of sediment and particulate phosphorus inputs into water bodies in the Swiss Midlands – A twenty-year field study of soil erosion, Catena, 203(March), 105290, doi:10.1016/j.catena.2021.105290, 2021.

Schmidt, S., Alewell, C. and Meusburger, K.: Mapping spatio-temporal dynamics of the cover and management factor (C-factor) for grasslands in Switzerland, Remote Sens. Environ., 211(April), 89–104, doi:10.1016/j.rse.2018.04.008, 2018.

---

## Referee Report (RR1)

**General comments**

I would like to thank the authors for revising the manuscript and for answering most my questions. The quality of the manuscript was improved largely by the revisions. However, some of my comments in the first review round were in my opinion not sufficiently addressed. I think the topic and scope of this manuscript are highly scientifically significant and I would like to see this paper published at some point. However, especially the answer to the second insufficiently addressed comment (regarding Figure 8) makes me question the scientific quality of the manuscript, and I think the manuscript still needs major revisions. However, I may be on my own with this judgement and I am looking forward to see the opinion of the other reviewers and of the editor.

Besides the insufficiently addressed comments from the first review round, I also have a few additional comments to the manuscript that are listed below. The line numbers relate to the author's tracked changes version of the manuscript.

**Insufficiently addressed comments from the first review round**

1) L251-252: I still strongly disagree with your statement that open source geodata of crop statistics are not available for the Baldegg catchment. This is simply wrong.
As mentioned in the previous review, these open source geodata are actually available in high resolution. They can be downloaded within few minutes using the link below and can be used for free (creative commons by license): https://geodienste.ch/services/lwb_nutzungsflaechen
(Additional information is provided here: https://daten.geo.lu.ch/produkt/lwnfmgdm_ref_v1)
Since in the dataset some of the agricultural areas (probably 10-20%) are missing a crop classification, you would still have to take some assumptions, but you would be able to represent the cropping reality much better.
The canton of Lucerne recently changed his law with regards to open data. Therefore it is correct that most probably this dataset was not completely open source by the time when you handed in the manuscript (e.g. for commercial use there was indeed a fee which had to be paid). However, I am pretty sure that the dataset was already available freely for governmental institutions and universities by that time.
Here, I must say that I have the feeling that, after being told about this dataset in the first revision, the authors did not really put a lot of effort into obtaining this dataset. I understand that using this dataset would mean a lot of additional effort for the authors, and that probably the added benefit to the manuscript would be small. However, it should be at least stated correctly in the manuscript that this dataset is available, but was not used due to reason XY.

2) L443-445: In the previous review round, I already asked the following question: *In the "Ron" stream, the 95% prediction interval seems much narrower than in the other rivers (Figure 8). Therefore, the observed values are mostly outside of this interval and the out of bound percentage is much higher than for the other streams. Can you explain this?*
Your answer was: *The interval is narrower because the model fit was better, and the residuals were lower. This led to a lower proportion of the observed values being outside the prediction interval.*
However, in Table 4, you report that the percentage of observations outside of the prediction interval was 61%, which is much higher than for the other catchments. This strongly contradicts your answer. In my opinion, there is something wrong with your fitted rating curve and you should fix this. (It is possible that I simply did not understand your answer correctly. In this case, I

would be very happy if you could explain it to me in a version "for dummies". Additionally, I would suggest that add a discussion to the manuscript regarding the influence of this high percentage of observations outside the prediction interval on your results.)

**Major comments**

L357: In Figure 5, the MSE increase for $CP_{arable}$ is missing. Additionally, the MSE increase reported for road connectivity does not correspond to the numbers in the text.

**Minor comments**

L133-134: What are these 10 degrees relating to? Average slope? Please clarify in the manuscript.

L249-250: Do you mean the dataset swissTLM3D (as referenced here) or the Swiss Map Vector 25 BETA (which you mention in your answer to the review)? In the first case, it does not make sense to me to mention the "1:25'000" scale, since – to my knowledge – there is no scale related to the swissTLM3D dataset. Additionally, to improve reproducibility of your work, you should in my opinion specify here that you first converted the roads to polygons by using buffers considering their widths.

L357: Explain the abbreviation RFA here. I know that you explain it in the text, but each Figure should be readable by itself.

L365: From what you write in the manuscript, I did not make the link to Table 3. This was only clear to me after your explanations to my questions. I therefore suggest to refer to Table 3 here.

L367: The combined use of hyphen and minus makes the legend of Figure 6 rather confusing. Please use another symbology for this (e.g. use "to" instead of the minus).

L420: Figure 7 is now much more convincing to me than it was in the previous version. Thanks for the adaptations.

L428-429: An additional question to your model came up when trying to understand why the model does represent the Ron catchment well. In L428-429 you write: "It is important to note that the median sediment concentrations calculated by the rating curve (Equation 1) underestimated the actual observations, for all tributaries." If your model aims on predicting mean annual sediment loads, should the median of your model not have a mean error of approximately zero?

L476-478: Thanks for clarifying which characteristics you analysed. You should not only mention that you determined the corresponding numbers for each catchment (stream density, road density, fraction covered by agricultural land/forest/infrastructure), but also provide them somewhere in the manuscript or in the supplementary information.
Furthermore, it is unclear to me what you mean exactly with land cover. Do you mean "(agricultural land, forests, and infrastructure (e.g. settlements, developed areas, and roads)", as you mention it in L113-114? In the next sentence, you are hypothesizing about the influence cropping specifities, which also relate to land cover. This makes this paragraph rather confusing. You should state more clearly, what you mean with "land cover" here.

**Technical corrections**

L147-149: Not sure if this sentence is grammatically correct.

L154: y-axis: Sediment Concentration → Sediment concentration

L157: Equation → equation

L162: $x_k$ → $x_{k,i}$

L165: The variable z (row 5) is not explained. Please explain.

L171: R package version?

L321-322: R package version?

---

## Referee Report (RR2)

**Referee report**

*A conceptual model-based sediment connectivity assessment for patchy agricultural catchments*

*hess-2021-231, submitted on 28 Apr 2021*

**General comments**

I would like to thank the authors for again revising the manuscript and for answering my questions. In my opinion, the quality of the manuscript was further improved.

Regarding the discussion on the narrower prediction interval of the Ron river and it's high out of bound percentage: After reading the explanations of the authors this part now somehow makes more sense to me. However, I must say that I still do not understand this part well enough to evaluate the quality of this part of the manuscript. The quality of this part should therefore be evaluated by the editor or another reviewer.

Besides this comment, I only have one other minor comment (see below). Apart from these comments, I am happy with the manuscript and I am looking forward to see it published.

**Minor comments**

L249: I still don't understand why you used the Swiss Map Vector 25 BETA here, but the swissTLM3D in L290 pp. Why are you using two different model versions for determining the Swiss land cover?

---

## Author Response (AR2)

Dear Genevieve Ali and referees,

Many thanks for another valuable round of reviews. We highly appreciated the opportunity to improve our work. We remain eager to address any further comments and suggestions.

First, we would like to state that we have uploaded the model input and output data, raw discharge and sediment concentration data, and the R scripts used for i) adjusting the sediment rating curves and propagating the regression uncertainty, ii) running the WaTEM/SEDEM model, iii) performing the random forest analysis (Figure 5), and iv) summarising model results (Figure 7) to the Zenodo data repository (https://doi.org/10.5281/zenodo.6560226). We hope the data, along with our responses and revisions, will address the concerns from Reviewer #2 regarding the scientific quality of our work.

Below we answer to all your comments. For clarity and improved visualisation, the editor and reviewer comments are shown from here on in black. The authors' replies are in blue font below each of the reviewers' statements. The changes in the revised manuscript are displayed in green. Line numbers refer to the revised tracked manuscript.

**Editor's comments:**

Dear Authors,

Two referees provided comments on your revised manuscript. One reviewer recommended that some technical corrections be made. The other reviewer was of the opinion that your manuscript was much improved, compared to the previous version, but they raised that some comments from the previous round of review had not been properly or sufficiently addressed. I do agree with their two main major comments:

1) It is not appropriate to state that data are not available if they are available (with respect to crop information, in your case). Unless the statement made to that effect in the revised manuscript was not properly phrased and therefore not purposefully misleading. If data are available in the right format, then it would be important to incorporate them into your manuscript. If some data are available but they do not meet a quality criterion (or a spatial coverage criterion or a spatial resolution criterion) that you are using, then it would be better to state so explicitly.

We completely agree it is inappropriate to state that data are not available if they are so. Our reasoning to state that open source geodata on crop information were not available within our study catchment was based on our interaction with the Swiss Geoservices portal on 09-10.08.2021. After the first review round, we contacted the geoportal to request access to the data mentioned by Reviewer #2 in their initial comments. However, as stated in our previous response letter, we were told the data were not freely available and access would cost 4,260 CHF (even though we mentioned the data would be used for research purposes). We can provide proof of this interaction with the Geoservices portal, if the reviewers

and the editor would like so, in order to clarify any misunderstandings. We had no intention to make any sort of misleading statements, and we apologise if that was somehow conveyed.

The parcel-specific crop data for the Baldegg catchment are indeed now freely available, as stated by Reviewer #2, and we were able to access it. As mentioned by Reviewer #2, canton Lucerne has changed their open data policy since our inquiry, and we were unaware of that. We are highly thankful to the reviewer for pointing this out.

However, as also mentioned by Reviewer #2, including such data to our manuscript at this point would be of little added value – mostly because i) we have dealt with the uncertainty in the landuse classification within our Monte Carlo simulation; ii) our aim with this paper is to investigate the effects of linear structures on sediment transfer and to explore connectivity scenarios, not to produce an ultrarealistic soil erosion map for the Baldegg catchment or to perform a rejectionist model test; and iii) although crop data from 2021 are available, we would still need to make a lot of assumptions for a model representing long-term (~20 years) average annual conditions, which would lead to very a similar parameter sampling approach to one we employed so far.

Hence, we have removed the sentence about data availability and explain our rationale as it follows:

"Due to the difficulties involved in accurately representing long-term average agricultural landuse patterns and farming management practices per field parcel, pastures and cropland were considered a single arable land category, using only the information available from the land cover map (Table 2) (Swisstopo, 2018). In this case, minimum and maximum values were relaxed to represent a wide possible combination of crops and support practices. Such combinations were assessed with the *CP*-Tool (Kupferschmied, 2019), which allows for the calculation of *CP* values considering common crop rotation systems in Switzerland. The minimum *CP* values were particularly reduced to include typical values for permanent grasslands in Switzerland (~0.01) (Schmidt et al., 2018b). This simplified approach should be appropriate considering i) our focus on connectivity scenarios and linear landscape structures, and ii) the use of the Monte Carlo simulation with the sampling of a wide parameter space that accounts for the uncertainty in the landuse classification". (L250-261)

We have also included additional information about the temporal scale represented by the model:

"The model is by default executed in an average yearly time step, as typical in RUSLE applications, which predict long-term (~20 years) average annual soil losses (…)" (L188-189).

"Suspended sediment loads are a product of a complex interaction of hillslope and channel remobilisation processes, which are not fully represented by WaTEM/SEDEM. In addition, since the model is RUSLE-based, the soil redistribution rates represent long-term average annual values, which hampers a straightforward comparison with annual sediment transport rates." (L331-333)

Since we have not used the parcel-specific crop data, we believe it would be confusing to the readers to mention such data in the manuscript. However, if the editor and the reviewers think it would be important to mention the availability of the parcel-specific crop data, we would of course be happy to do so.

2) Some of the comments made by one reviewer regarding the results reported in Figure 8, Table 4 and Figure 5 need to be addressed/discussed in the manuscript, because other readers may raise similar questions (personally, I know that, I, too did not understand how a better model fit could lead to a larger percentage of out-of-bound observations).

We apologise for the blunter in Figure 5 and for not providing a more thorough response regarding Table 4/ Figure 8. We have corrected the mistake in Figure 5, provided a detailed explanation to the reviewer's question, and addressed the topic of the rating curves and the percentage of out-of-bound observations in the text. Please see our response to Reviewer #2 below.

**Reviewer #1**

We are highly thankful for another round of thoughtful comments. We have incorporated your suggestions to the manuscript and replied to all your questions below.

L 45: Not to mention transport of phosphorus, as you point out below.

Thank you for pointing this out, we have included a mention phosphorus delivery to surface waters (L 42).

L 49: Is the transport to downstream systems not considered part of sediment connectivity? I guess it would depend on the defined spatial and temporal scale.

Indeed, we believe this depends on the spatial and temporal scale of the analysis, and perhaps on how one interprets the definition from Heckman et al. (2018). But in general, we understand that sediment connectivity research is more focused on internal sediment transport processes than on catchment sediment yields, or sediment export to downstream systems.

L 149: A bit more information on the inclusion of a hysteresis parameter could be included here - Is this referring to hysteresis in the sense of Sheriff et al. 2016? Sherriff, S. C., Rowan, J. S., Fenton, O., Jordan, P., Melland, A. R., Mellander, P. E., & Huallachain, D. O. (2016). Storm event suspended sediment-discharge hysteresis and controls in agricultural watersheds: implications for watershed scale sediment management. Environmental Science & Technology, 50(4), 1769-1778.

Thank you, we agree we could have included some more information regarding hysteresis and seasonality in sediment export patters and rating curves. Following your comment, we made the following adjustments "The rating curve partially accounts for hysteresis and seasonality (Table 1), which can have a significant impact on sediment export patterns and reflect the catchment landuse, hydrological connectivity, and internal sediment source dynamics (Sherriff et al., 2016). To derive the parameters in equation 1 we used a parsimonious multivariate regression which does not require separate calibration for different seasons (Cohn et al., 1992; Vigiak and Bende-Michl, 2013)." (L149-153)

L 230: It might be useful to state why LS isn't varied – I'm guessing there isn't too much uncertainty associated with the DEM or the LS equation?

Apologies for the missing information and thanks for raising this question. We do believe the LS equation is a source of uncertainty, although a lesser one compared to the CP and K factors (see Batista et al., 2021). However, since we used a high-resolution DEM, the variance associated to the LS factor is much reduced, compared to the typical set up in which soil redistribution models are applied. We included an explanation to this matter in the text: "Finally, the $LS_{2d}$ factor was calculated with a slope (rad) and an upslope contributing area (m$^2$) grid, which were obtained by processing a 2 m x 2 m resolution DEM from SwissALTI3D (Swisstopo, 2014a). In this case, the error in the $LS_{2d}$ factor was

not incorporated into Monte Carlo simulation due to the use of the high-resolution DEM, which should considerably reduce the uncertainty associated to the parameter estimation." (L261-265)

L 417 – 473: this is a bit confusing. Is there a typo here?

Thank you for noticing this. We corrected to: "In a wider context, our study has demonstrated how structural sediment connectivity patterns can be investigated with a conceptual model such as WaTEM/SEDEM, provided that model spatial resolution is sufficiently fine to represent relevant features and processes." (L503-505)

**Reviewer #2**

**General comments**

I would like to thank the authors for revising the manuscript and for answering most my questions. The quality of the manuscript was improved largely by the revisions. However, some of my comments in the first review round were in my opinion not sufficiently addressed. I think the topic and scope of this manuscript are highly scientifically significant and I would like to see this paper published at some point. However, especially the answer to the second insufficiently addressed comment (regarding Figure 8) makes me question the scientific quality of the manuscript, and I think the manuscript still needs major revisions. However, I may be on my own with this judgement and I am looking forward to see the opinion of the other reviewers and of the editor.

Thank you for another thorough review of our manuscript, we highly appreciate it. We are sorry our replies from the previous review round were did not sufficiently address your concerns. As stated above, we have uploaded all our data and code to the Zenodo data repository, making it fully transparent and reproducible. We hope this, in combination with our responses below, will convince you of the scientific quality of our work. We highly welcome the constructive criticism and remain eager to address any points you find necessary. Below we provide more detailed responses to your comments.

**Insufficiently addressed comments from the first review round**

1) L251-252: I still strongly disagree with your statement that open source geodata of crop statistics are not available for the Baldegg catchment. This is simply wrong. As mentioned in the previous review, these open source geodata are actually available in high resolution. They can be downloaded within few minutes using the link below and can be used for free (creative commons by license): https://geodienste.ch/services/lwb_nutzungsflaechen (Additional information is provided here: https://daten.geo.lu.ch/produkt/lwnfmgdm_ref_v1) Since in the dataset some of the agricultural areas (probably 10-20%) are missing a crop classification, you would still have to take some assumptions, but you would be able to represent the cropping reality much better.

The canton of Lucerne recently changed his law with regards to open data. Therefore it is correct that most probably this dataset was not completely open source by the time when you handed in the manuscript (e.g. for commercial use there was indeed a fee which had to be paid). However, I am pretty sure that the dataset was already available freely for governmental institutions and universities by that time. Here, I must say that I have the feeling that, after being told about this dataset in the first revision, the authors did not really put a lot of effort into obtaining this dataset. I understand that using this dataset would mean a lot of additional effort for the authors, and that probably the added benefit to the manuscript would be small. However, it should be at least stated correctly in the manuscript that this dataset is available, but was not used due to reason XY.

We apologise if we gave the impression that we did not put effort into accessing the crop data. Following your comments in the previous review round, we contacted the Swiss Geoservices portal on 09.08.2021, requesting access to the "Landw. Bewirtschaftung: Nutzungsflächen" data for the Baldegg catchment, for scientific purposes and for the University of Basel. We received a reply on 10.08.2021, stating that the data would cost 4,260 CHF, unless the research was being done on behalf of or in cooperation with Canton Lucerne. This was our reasoning for stating in the revised manuscript that open source geodata on crop statistics were not available for the study area, as this was indeed the case at the time. We can provide proof of these interactions with the Geoservices portal, if the reviewer and the editor would like so, in order to clarify any misunderstandings.

Indeed, the crop data is now freely available and we were able to access it – many thanks for pointing this out. As you mentioned, however, including this dataset to our analysis at this point would have little added value to the manuscript, as we already dealt with the uncertainty in the land use classification with the Monte Carlo simulation, in which we sampled parameter values from both arable land (considering different crops) and grasslands. In addition, our aim here is not to provide an ultrarealistic soil erosion map for the Baldegg catchment, but whether to explore connectivity scenarios and investigate the influence of linear landscape structures on sediment transfer. Furthermore, although crop data from 2021 are available, we would still need to make a lot of assumptions considering crop rotation (including temporary pastures) for a model representing average annual conditions, which would lead to very a similar parameter sampling approach to one we employed so far.

Hence, we have removed the sentence about data availability and explain our rationale as it follows:

"Due to the difficulties involved in accurately representing long-term average agricultural landuse patterns and farming management practices per field parcel, pastures and cropland were considered a single arable land category, using only the information available from the land cover map (Table 2) (Swisstopo, 2018). In this case, minimum and maximum values were relaxed to represent a wide possible combination of crops and support practices. Such combinations were assessed with the *CP*-Tool (Kupferschmied, 2019), which allows for the calculation of *CP* values considering common crop rotation systems in Switzerland. The minimum *CP* values were particularly reduced to include typical values for permanent grasslands in Switzerland (~0.01) (Schmidt et al., 2018b). This simplified approach should be appropriate considering i) our focus on connectivity scenarios and linear landscape structures, and ii) the use of the Monte Carlo simulation with the sampling of a wide parameter space that accounts for the uncertainty in the landuse classification." (L250-261)

We have also included additional information about the temporal scale represented by the model:

"The model is by default executed in an average yearly time step, as typical in RUSLE applications, which predict long-term (~20 years) average annual soil losses (…)" (L186-187).

"Suspended sediment loads are a product of a complex interaction of hillslope and channel remobilisation processes, which are not fully represented by WaTEM/SEDEM. In addition, since the model is RUSLE-based, the soil redistribution rates represent long-term average annual values, which hampers a straightforward comparison with annual sediment transport rates." (L327-331).

Since we have not used the parcel-specific crop data, we believe it would be confusing to the reader to mention such data. However, if the editor and the reviewers think it would be important to mention the availability of this data, we would of course be happy to do so.

2) L443-445: In the previous review round, I already asked the following question: In the "Ron" stream, the 95% prediction interval seems much narrower than in the other rivers (Figure 8). Therefore, the observed values are mostly outside of this interval and the out of bound percentage is much higher than for the other streams. Can you explain this? Your answer was: The interval is narrower because the model fit was better, and the residuals were lower. This led to a lower proportion of the observed values being outside the prediction interval.

However, in Table 4, you report that the percentage of observations outside of the prediction interval was 61%, which is much higher than for the other catchments. This strongly contradicts your answer. In my opinion, there is something wrong with your fitted rating curve and you should fix this. (It is possible that I simply did not understand your answer correctly. In this case, I would be very happy if you could explain it to me in a version "for dummies". Additionally, I would suggest that add a discussion to the manuscript regarding the influence of this high percentage of observations outside the prediction interval on your results.)

We agree we could have given a more detailed answer to this question, and we are sorry you had to ask again. In all honesty, perhaps we did not give much attention to this question because the sediment load estimates were a somewhat secondary topic in the manuscript. Your questions made us realise that the way the manuscript was drafted put too much focus on the sediment rating curve and the uncertainty estimation of the sediment loads, which in the end created confusion and deviated the reader from our main findings. That is, we only included the sediment load estimates (with a quantification of the regression uncertainty) to provide a generable idea of the plausibility of the model simulations, as we explain in the manuscript. Therefore, below we answer to your questions in detail and explain how we edited the manuscript to improve the clarity of our paper.

First, we would like to point out that we have re-calculated all the sediment rating curves and sediment loads in order to be completely sure there was nothing wrong with the estimates. Our curve fitting and error propagation calculations were correct, but we decided to remove the covariate number 5 from the sediment rating curves, as upon reflection following one of your minor comments, we realised that we ourselves were having trouble understanding the reasoning behind the parameter $z$ in the covariate equation. We had used a value of 15 days to consider short-term exhaustion, as we understood was

suggested in Vigiak and Bende-Michl (2013), but in the end we were not happy making this assumption as we were not completely confident regarding the correctness of our interpretation. As you see, the exclusion of covariate number 5 led to small changes in the uncertainty bands from Figure 7 and in some of the evaluation metrics in Table 4. Please note that the differences in the estimated average annual sediment loads, which we compared to model outputs, were minimal. Although some of the evaluation metrics showed a decrease for some streams, we preferred to stick to the covariates we were completely satisfied with and that would not add unnecessary confusion to the manuscript.

Now, regarding your question: the approach we used to quantify the uncertainty in the sediment rating curves is based on the residual errors from the regression models displayed in equation 1/table 1. Once the models were adjusted, we used the *sim* function from the R package *arm* to draw posterior simulations of the model coefficients (the intercept and the coefficients). If the residuals from the initial model fit were large (poor fit), then the posterior distribution of model coefficients was broad. Similarly, if the initial model fit had low residuals, the spread of the posteriors was narrow.

For example, see below the quantile values for the posteriors of the model coefficients for the rating curves for the Mulibach and the Ron. The sediment rating curve for the Mulibach had a poorer fit (Residual Standard Error = 1.2, $R^2$ = 0.52) compared to the Ron (Residual Standard Error = 0.90, $R^2$ = 0.60) and hence the posterior distributions of the model coefficients were much wider (in particular for the intercept and $x_1$).

| Stream | Quantile | Coefficients | | | | |
| --- | --- | --- | --- | --- | --- | --- |
| | | (Intercept) | $x_1$ | $x_2$ | $x_3$ | $x_4$ |
| Mulibach | 0% | 8.26 | 2.01 | 0.14 | -0.73 | -0.59 |
| Mulibach | 25% | 9.12 | 2.49 | 0.21 | -0.51 | -0.48 |
| Mulibach | 50% | 9.42 | 2.64 | 0.23 | -0.44 | -0.42 |
| Mulibach | 75% | 9.77 | 2.85 | 0.25 | -0.37 | -0.34 |
| Mulibach | 100% | 10.64 | 3.52 | 0.36 | -0.20 | -0.09 |
| Ron | 0% | 3.52 | 0.91 | 0.08 | -0.38 | -0.42 |
| Ron | 25% | 3.71 | 1.06 | 0.14 | -0.22 | -0.26 |
| Ron | 50% | 3.75 | 1.11 | 0.16 | -0.16 | -0.21 |
| Ron | 75% | 3.78 | 1.14 | 0.19 | -0.13 | -0.15 |
| Ron | 100% | 3.91 | 1.24 | 0.27 | 0.04 | -0.03 |

If we take the range from the posteriors of model coefficients to provide a range of model realisations, then the regression models with greater residuals and poorer fits will have larger uncertainty bands and the other way around. If the uncertainty bands from the model realisations are very wide, they might encompass a larger number of observations, even if the regression model had poorer fit and the

predictions from the median of the realisations showed a higher RMSE. Hence, one could understand this a trade-off between accuracy and precision, as seen for instance in the example from the rating curves for the Ron and the Mülibach, and their respective proportion of observations not encompassed by the uncertainty bands, as well as their RMSE, ME, and NSE values (Table 4).

We have now addressed this topic in the text, starting in the methods:

"In order to estimate continuous daily sediment concentration values, later used to produce average yearly sediment loads for each tributary, we used a rating curve approach (Equation 1), combining the roughly triweekly sediment concentration measurements with continuous discharge measurements. The rating curve partially accounts for hysteresis and seasonality (Table 1), which can have a significant impact on sediment export patterns and reflect the catchment landuse, hydrological connectivity, and internal sediment source dynamics (Sherriff et al., 2016). To derive the coefficients in equation 1 we used a parsimonious multivariate regression which does not require separate calibration for different seasons (Cohn et al., 1992; Vigiak and Bende-Michl, 2013)." (L146-154)

"To analyse the uncertainty in the regressions we simulated posterior distributions of the model coefficients ($\beta_0$, $\beta_k$) with an informal Bayesian function of the R package 'arm' (Gelman and Hill, 2007), as in Batista et al. (2021). This function produces realisations of model coefficients based on the residual standard error of the regression, which means that models with poorer fits will yield broader posterior distributions of regression coefficients. The posterior distributions were used to simulate 1000 sediment concentration values for each day $i$. These were transformed into daily distributions of sediment loads (Mg), considering the mean daily discharge measurements from the gauging stations. Sediment loads were ultimately aggregated into average annual values (Mg yr$^{-1}$) with uncertainty bands, which should allow for a general comparison with the different sediment connectivity scenarios simulated by WaTEM/SEDEM." (L165-175)

Moreover, we addressed the observations out-of-bound from the regression uncertainty bands in the results. In this case, we moved Figure 8 to the supplementary material, in order to avoid an excessive focus on the sediment rating curves. We think this will improve the readability of the paper and we appreciate how your comments prompted these changes.

"It is important to note that the median daily sediment concentrations calculated from the 1000 realisations of the rating curves (Equation 1) underestimated the high sediment concentration measurements, for all tributaries. This resulted in the positive mean error of the median estimates (Table 4). Moreover, the Nash-Sutcliffe model efficiency coefficient for the median calculations was unsatisfactory considering the usual thresholds for model performance (e.g. Moriasi et al., 2015). However, the 95 % prediction interval of the rating curves encompassed a large proportion of the sediment concentration observations for the tributaries with poorer fits and wider uncertainty bands (i.e., the Höhibach, Mülibach, and Spittlisbach) (Table 4, Supplementary Material Figure 1). The sediment

rating curves for the tributaries which displayed a better fit (i.e., the Ron and Stägbach) encompassed a much lesser proportion of the observed sediment concentration values (Table 4, Supplementary Material Figure 1). That is, the regressions with the lowest residual standard errors had narrower uncertainty bands, which albeit produced more accurate median predictions, led to a greater proportion of observations out-of-bound from the 95 % prediction interval. In any case, the largest errors were associated to underestimates of extreme events, and therefore, it is likely that actual sediment loads from the tributaries are contained within the long right side of the skewed distributions resulting from the error propagation of the rating curves (Figure 7), which would increase the overlap with the shortcut scenario." (L425-442)

**Major comments**

L357: In Figure 5, the MSE increase for CParable is missing. Additionally, the MSE increase reported for road connectivity does not correspond to the numbers in the text.

Apologies for this blunter in Figure 5. The label for the $CP_{arable}$ was missing and the Road Connectivity label was misplaced, thus the divergence from the numbers in the text. This was probably caused by re-ordering the factors in the facet plots. Thanks for picking this up, we completely missed it!

In all honesty the first author could not find the last piece of code he used for this figure, which led to again some new numbers in the random forest analysis importance ranking, due to the randomness of the tree building and because we increased the number of trees (L351-356). This did not change the order of the importance ranking or the interpretation of the analysis. As we mentioned above, the code for reproducing the analysis and the figure is now available, with a set seed to insure the same results in case of another run of the random forest.

**Minor comments**

L133-134: What are these 10 degrees relating to? Average slope? Please clarify in the manuscript.

Apologies for the missing information. We updated the information in the manuscript to: "Steeper slopes (average values above 10°) (…)". (L123)

L249-250: Do you mean the dataset swissTLM3D (as referenced here) or the Swiss Map Vector 25 BETA (which you mention in your answer to the review)? In the first case, it does not make sense to me to mention the "1:25'000" scale, since – to my knowledge – there is no scale related to the swissTLM3D dataset. Additionally, to improve reproducibility of your work, you should in my opinion specify here that you first converted the roads to polygons by using buffers considering their widths.

Apologies again with the confusion regarding the Swisstopo references. Indeed, the data for the roads was retrieved from the swissTLM3D (Swisstopo 2020), and then converted to polygons using a buffer. This polygon merged with the Swiss Map Vector 25 BETA (Swisstopo, 2018) and rasterised to produce

a land cover grid which included roads, tree lines and hedges. We made these corrections and included the information you requested in the text:

"For the $C$ and $P$ factors, here combined in a single $CP$ parameter, uniform distributions were created for each landuse class in the catchment, based on commonly used values from the literature and a land cover map (1:25000) (Swiss Map Vector 25 BETA) (Swisstopo, 2018), which we rasterised to the model resolution (2 m x 2 m)." (L246-248)

"Hedges and tree lines within field blocks were already classified in the large-scale topographic landscape model of Switzerland (swissTLM3D) (Swisstopo, 2020) and required no additional processing apart from a merge with the land cover map (Swiss Map Vector 25 BETA) (Swisstopo, 2018). Furthermore, three road connectivity assumptions were assessed for each model iteration. For such, we first converted the roads from polylines (as available in the swissTLM3D) to polygons, using a buffer distance based on the road widths. Next, these polygons were rasterised and incorporated into the land cover grid used for modelling." (L289-296)

Finally, we included the correct citation for the Swiss Map Vector 25 BETA to the reference list:

Swisstopo. Swiss Map Vector 25 Beta, Das digitale Landschaftsmodell der Schweiz. 2018.

L357: Explain the abbreviation RFA here. I know that you explain it in the text, but each Figure should be readable by itself.

Thanks for noticing this, we spelled out Random Forest Analysis (RFA) in the figure caption:

"Figure 5. Mean squared error (MSE) increase associated to model input factors for the Random Forest Analysis (RFA). Larger relative errors indicate the input factors were more important for estimating model outputs." (L359-361)

L365: From what you write in the manuscript, I did not make the link to Table 3. This was only clear to me after your explanations to my questions. I therefore suggest to refer to Table 3 here.

Thank you, we now refer to Table 3 (L370).

L367: The combined use of hyphen and minus makes the legend of Figure 6 rather confusing. Please use another symbology for this (e.g. use "to" instead of the minus).

Thank you for the suggestion, we changed the symbology in the legend from Figure 6 using "to" instead of minus.

L420: Figure 7 is now much more convincing to me than it was in the previous version. Thanks for the adaptations.

Thank you for suggesting these adaptations, we agree this is more convincing.

L428-429: An additional question to your model came up when trying to understand why the model does represent the Ron catchment well. In L428-429 you write: "It is important to note that the median sediment concentrations calculated by the rating curve (Equation 1) underestimated the actual observations, for all tributaries." If your model aims on predicting mean annual sediment loads, should the median of your model not have a mean error of approximately zero?

Again, we apologise, as we could have been more precise regarding these statements. The regression model is used to estimate daily sediment concentrations, which are then used to calculate the sediment loads, in combination with the daily water discharge measurements (please see our response to your previous question about the rating curve above).

An unbiased model should indeed have mean error of approximately zero, but, as we now highlighted in in the text, the regression models particularly underpredicted the high sediment concentrations (L423-424, L435-436). Hence, there is positive mean error (observations higher than predictions), even for the median of the realisations of the rating curves. This can be also visualised in the figure below (solid line is the perfect fit), which we prepared for the reviewer:

[Figure]

L476-478: Thanks for clarifying which characteristics you analysed. You should not only mention that you determined the corresponding numbers for each catchment (stream density, road density, fraction covered by agricultural land/forest/infrastructure), but also provide them somewhere in the manuscript or in the supplementary information. Furthermore, it is unclear to me what you mean exactly with land cover. Do you mean "(agricultural land, forests, and infrastructure (e.g. settlements, developed areas, and roads)", as you mention it in L113-114? In the next sentence, you are hypothesizing about the

influence cropping specifities, which also relate to land cover. This makes this paragraph rather confusing. You should state more clearly, what you mean with "land cover" here.

Apologies for not including such information beforehand. We supplied the characteristics of the sub-catchments in the supplementary material:

Supplementary Material Table 1. Land cover and physiographic characteristics of the analysed sub-catchments draining into the Lake Baldegg.

| Attribute | | Sub-catchment | | | | |
|---|---|---|---|---|---|---|
| | | Höhibach | Mülibach | Ron | Spitllisbach | Stägibach |
| Land cover (%) | Arable land | 72 | 56 | 71 | 52 | 76 |
| | Forest | 13 | 40 | 12 | 25 | 8 |
| | Infrastructure | 4 | 3 | 12 | 11 | 10 |
| | Orchards | 11 | 1 | 4 | 12 | 6 |
| | Other* | 0 | 0 | 1 | 0 | 0 |
| Road density (km km$^{-2}$) | | 5.9 | 6.0 | 6.9 | 8.4 | 6.7 |
| Stream density (km km$^{-2}$) | | 2.0 | 3.0 | 1.9 | 3.0 | 1.2 |
| Mean elevation (m a. s. l.) | | 617 | 708 | 568 | 634 | 545 |
| Mean slope (%) | | 12 | 16 | 9 | 12 | 9 |

* Water surfaces and rock outcrops.

We refer to the table in the text and explain the following: "Of note, the contrasting results for the Höhibach sediment loads (Figure 7), which are much closer to the sink and patch-connector simulations, do not seem to be explained by any physiographical characteristic of the sub-catchment (Supplementary Material Table 1). Hence, we speculate that this different pattern could be caused by a lower inlet drainage density or specific farming practices within the Höhibach contributing area." (L480-485)

**Technical corrections**

L147-149: Not sure if this sentence is grammatically correct.

We rephrased to: "On average 274 grab samples were taken from each tributary, which corresponds to one sample every 22 days, in addition to the samples collected during high-flow events (10 – 13 per year) (Figure 3)". (L136-138)

L154: y-axis: Sediment Concentration -> Sediment concentration

We changed the y-axis title in Figure 3 to "Sediment concentration".

L157: Equation -> equation

In this case we kept the capital letter, as in the rest of the manuscript.

L162: xk -> xk,i

Updated to $x_{k,i}$ (L155).

L165: The variable z (row 5) is not explained. Please explain.

As explained above, we removed covariate number 5 from the rating curve.

L171: R package version? L321-322: R package version?

Sorry we did not add these package versions before. They are now available in the references:

Gelman, A. and Hill, J.: Data Analysis Using Regression and Multilevel/Hierarchical Models, Cambridge Univeristy Press, New York., R package version 1.12.2, 2007.

Liaw, A., Wiener, M.: Classification and regression by randomForest. R News, 2, 18–22, R package version 4.7.1, 2002.

**References**

Batista, P. V. G., Laceby, J. P., Davies, J., Carvalho, T. S., Tassinari, D., Silva, M. L. N., Curi, N. and Quinton, J. N.: A framework for testing large-scale distributed soil erosion and sediment delivery models: Dealing with uncertainty in models and the observational data, Environ. Model. Softw., 137, doi:10.1016/j.envsoft.2021.104961, 2021.

Heckmann, T., Cavalli, M., Cerdan, O., Foerster, S., Javaux, M., Lode, E., Smetanová, A., Vericat, D. and Brardinoni, F.: Indices of sediment connectivity: opportunities, challenges and limitations, Earth-Science Rev., 187(December 2017), 77–108, doi:10.1016/j.earscirev.2018.08.004, 2018.

Vigiak, O. and Bende-Michl, U.: Estimating bootstrap and Bayesian prediction intervals for constituent load rating curves, Water Resour. Res., 49(12), 8565–8578, doi:10.1002/2013WR013559, 2013.

---

## Author Response (AR3)

Dear Genevieve Ali,

Thank you for the editing work throughout this review process.

Regarding the reviewer's last question, we used the two land cover products because they provide complementary information. This is now explained in lines 283-287:

"Hedges and tree lines within field blocks were already classified in the large-scale topographic landscape model of Switzerland (swissTLM3D) (Swisstopo, 2020) and required no additional processing apart from a merge with the land cover map (Swiss Map Vector 25 BETA) (Swisstopo, 2018). These two land surface models were combined since they contain differently assigned and complementary land cover object classes."

With thanks and best wishes,

Pedro Batista